# Simulation of the future sea level contribution of Greenland with a new glacial system model

Reinhard Calov[1], Sebastian Beyer[1,2,3], Ralf Greve[4], Johanna Beckmann[1], Matteo Willeit[1], Thomas Kleiner[2], Martin Rückamp[2], Angelika Humbert[2,3], and Andrey Ganopolski[1]

[1]Potsdam Institute for Climate Impact Research, Earth System Analysis, Potsdam, Germany
[2]Alfred Wegener Institute, Helmholtz Centre for Polar and Marine Research, Bremerhaven, Germany
[3]University of Bremen, Bremen, Germany
[4]Institute of Low Temperature Science, Hokkaido University, Sapporo, Japan

*Correspondence to:* Reinhard Calov (calov@pik-potsdam.de)

**Abstract.** We introduce the coupled model of the Greenland glacial system IGLOO 1.0, including the polythermal ice sheet model SICOPOLIS (version 3.3) with hybrid dynamics, the model of basal hydrology HYDRO and a parameterization of submarine melt for marine-terminated outlet glaciers. Aim of this glacial system model is to gain a better understanding of the processes important for the future contribution of the Greenland ice sheet to sea level rise under future climate change scenarios. The ice sheet is initialized via a relaxation towards observed surface elevation, imposing the palaeo-surface temperature over the last glacial cycle. As a present-day reference, we use the 1961–1990 standard climatology derived from simulations of the regional atmosphere model MAR with ERA reanalysis boundary conditions. For the palaeo-part of the spin-up, we add the temperature anomaly derived from the GRIP ice core to the years 1961–1990 average surface temperature field. For our projections, we apply surface temperature and surface mass balance anomalies derived from RCP 4.5 and RCP 8.5 scenarios created by MAR with boundary conditions from simulations with three CMIP5 models. The hybrid ice sheet model is fully coupled with the model of basal hydrology. With this model and the MAR scenarios, we perform simulations to estimate the contribution of the Greenland ice sheet to future sea level rise until the end of the 21st and 23rd centuries. Further on, the impact of elevation–surface mass balance feedback, introduced via the MAR data, on future sea level rise is inspected. In our projections, we found the Greenland ice sheet to contribute to global sea level rise between 1.9 and 13.0 cm until the year 2100 and between 3.5 and 76.4 cm until the year 2300, including our simulated additional sea level rise due to elevation–surface mass balance feedback. Translated into additional sea level rise, the strength of this feedback in the year 2100 varies from 0.4 to 1.7 cm, and in the year 2300 it ranges from 1.7 to 21.8 cm. Additionally, taking Helheim and Store Glaciers as examples, we investigate the role of ocean warming and surface runoff change for the melting of outlet glaciers. It shows that ocean temperature and subglacial discharge are about equally important for the melting of the examined outlet glaciers.

## 1   Introduction

Since the last decade of the 20th century, the Greenland ice sheet (GrIS) loses mass with accelerating speed (e. g. Helm et al., 2014; Talpe et al., 2017), shaping one of the most important contributors to sea level rise (Shepherd et al., 2012; Rietbroek

et al., 2016; Forsberg et al., 2017). This mass loss is not only driven by decreasing surface mass balance (SMB), but also by increasing ice discharge via outlet glaciers. The partition between these two contributions to GrIS mass loss is about equal (Rignot et al., 2011; Box and Colgan, 2013; Enderlin et al., 2014; van den Broeke et al., 2016). Understanding the processes determining the GrIS ice loss is vital for estimates of its contribution to future sea level rise.

Nowadays, the scientific community recognizes the large Greenland island as a complex system mainly composed of the ice sheet and numerous outlet glaciers (Joughin et al., 2010; Rignot and Mouginot, 2012), in subtle interaction with the surrounding ocean via fjord circulation (Straneo et al., 2012; Murray et al., 2010), and uprising meltwater plumes in an interplay with the calving outlet glaciers (O'Leary and Christoffersen, 2013). In our paper, we introduce the model IGLOO (Ice sheet model for Greenland including Ocean and Outlet glaciers, Fig. 1) intended to represent the major processes important for the future mass

changes of the GrIS on timescales of some centuries. The contribution to future sea level rise of the several glaciers and ice caps detached from the GrIS is small compared to the ice sheet and its attached outlet glaciers (Forsberg et al., 2017). Anyhow, these glacial bodies are treated by SICOPOLIS in our approach. The idea of IGLOO is to capture the complexity of the system by its involved model components and, at the same time, to treat the description of all single components as detailed as necessary (Claussen et al., 2002). We aim to have a tool with sufficient computational efficiency to enable large ensemble simulations on

timescales important for future climate change.

Knowledge of the present-day state of the GrIS has been improving considerably. Not only that there are reliable data from numerous observations (e. g. Velicogna and Wahr, 2005; Bales et al., 2009; Morlighem et al., 2014), but also several modelling studies exist. Present-day GrIS velocities are resolved by ice sheet models in horizontal resolutions as high as 600 m, including flow patterns of outlet glaciers (Aschwanden et al., 2016). Robinson et al. (2012) explicitly demonstrated the

multistable-hysteresis behaviour of the GrIS with a threshold of 1.6 °C above present-day global temperature for the decay of the GrIS; although such a decay will last at least about 1000 years. The past climate is an important element for GrIS ice sheet modelling as well, as it serves as a constraint for parameters particularly capturing the present-day GrIS (Robinson et al., 2011; Stone et al., 2013), and as it provides the history of the temperature field inside the present-day GrIS (Goelzer et al., 2013), which is important for the initialization of the GrIS in future warming simulations. However, palaeo-simulations with

free surface have the drawback that their resulting present-day ice thickness can differ considerably from observations (e. g. Calov et al., 2015). Such a simulated ice thickness is an unfavourable initial condition for projections because, in this case, the future simulation would start with ice which resides at the wrong locations or is absent at positions where it should reside according to observations. This leads to an erroneous drift in projected ice volume evolution. Therefore, we opt for a fixed domain approach (Calov and Hutter, 1996) in our palaeo-spin-ups or, more precisely, for a scheme that relaxes the simulated

surface elevation towards the observed one (Aschwanden et al., 2013). This approach has the advantage that it provides a good approximation of the present-day temperature-velocity field for initialization and at the same time prevents a spurious response in volume during future simulations of several hundred years. Different initialization methods are discussed by Saito et al. (2016).

There are several approaches to project future ice mass change of the GrIS, often with a special focus on a certain component

of the Greenland glacial system. Classical SMB approaches assume a passive ice sheet, but resolve the atmosphere with general

circulation models of the atmosphere (e. g. Gregory and Huybrechts, 2006) or additionally with a regional model (van Angelen et al., 2012; Rae et al., 2012; Fettweis et al., 2013). Several pioneering studies used three-dimensional dynamic ice sheet models in the shallow ice approximation (SIA) for projections of GrIS sea level contribution (e. g. Huybrechts and de Wolde, 1999; Greve, 2000). Later, higher-order (Fürst et al., 2013) or even full-Stokes (Gillet-Chaulet et al., 2012; Seddik et al., 2012)

ice dynamics was included for GrIS future projections. In a higher-order ice sheet model, Fürst et al. (2015) parameterizes ice sliding via ocean-temperature rise due to future climate change to investigate the impact of ocean warming on future projections of GrIS sea level contribution. Studies with an atmosphere-ocean general circulation model coupled to a SIA ice sheet model via surface-energy fluxes were undertaken by Vizcaino et al. (2015). Inspections of GrIS sea level contribution with a special focus on outlet glaciers were accomplished with a 3-D ice sheet model by Peano et al. (2017) or with a 1-D shallow shelf

model (Nick et al., 2013).

Here, we opt for the new version of SICOPOLIS v3.3 (Bernales et al., 2017). This version includes hybrid dynamics, which incorporates via the shelfy stream approximation (SStA; MacAyeal, 1989) longitudinal and lateral stresses, which are important for nearer-margin fast flow areas, along with horizontal plane shear (Hindmarsh, 2004) via the shallow ice approximation (SIA), important for the slow-flow regions in the more central regions of the ice sheet. Hybrid models have been developed before

by Pollard and DeConto (2007, 2012); Bueler and Brown (2009); Hubbard et al. (2009); Winkelmann et al. (2011); Fürst et al. (2013); Pattyn (2017). They are a compromise between the shallow ice approximation and the full-Stokes approach. Key of these hybrid models is that SIA and SStA operate on a common domain, although there are other approaches to treat longitudinal and lateral stresses (Ritz et al., 2001). Compared to the SIA, the hybrid dynamics is more promising in reproducing the velocity field of the GrIS in the catchment area of ice streams, where there is already fast flow (Rignot and Mouginot, 2012).

We do not employ ice-shelf dynamics in SICOPOLIS because the dynamics of outlet glaciers, which can have a floating ice tongue, is part of the outlet glacier component of IGLOO. We investigate the response of GrIS outlet glaciers to global warming (including ocean warming) with IGLOO in a separate paper (Beckmann et al., 2018a).

Models assuming a basal water layer for treatment of subglacial hydrology (Shreve, 1972) were often applied to the Antarctic ice sheet (Le Brocq et al., 2009; Kleiner and Humbert, 2014). Here, we apply such a model to the GrIS, because it captures

the major pathways of basal water toward the outlet glaciers (Livingstone et al., 2013), i. e. the model resolves in a good approximation the partition of basal water for the main GrIS outlet glaciers. This is important for reproducing the subglacial discharge of outlet glaciers, which is fed into our model of meltwater plumes. Further on, our model for basal hydrology simulates a thickening of the basal water layer toward the major GrIS outlet glaciers, regions over which the ice velocity becomes higher (Rignot and Mouginot, 2012). Therefore, we couple the ice velocities to the basal water layer, while the basal

melt rate of the ice sheet model provides the input to the model of basal hydrology. We expect this approach to be suitable for large-scale modelling of ice sheets on decadal timescales.

Simulating submarine melt rates at tidewater glaciers has been accomplished with different models that all share the core of the buoyant-plume theory (Sciascia et al., 2013; Xu et al., 2013; Slater et al., 2015; Carroll et al., 2015; Cowton et al., 2015; Slater et al., 2017). Recent studies (Jackson et al., 2017; Beckmann et al., 2018b) show that the line plume model by

Jenkins (2011) is an adequate tool to determine submarine melt rates for tidewater glaciers. In our paper, we apply the recently

developed line plume model by Beckmann et al. (2018b), based on the equations by Jenkins (2011), to two outlet glaciers, Store and Helheim Glaciers (Fig. 2), of the Greenland ice sheet. We have chosen Helheim and Store Glaciers for investigating the impact of future warming on glacier melt and for testing our methods because they are well examined glaciers. Numerous studies on these glaciers and their connecting fjord systems to the open ocean exist (Straneo et al., 2011; Sutherland and Straneo, 2012; Rignot et al., 2015; Jackson et al., 2014; Chauché et al., 2014). Some provide data on temperature- and salinity profiles inside the fjord from conductivity-temperature-depth (CTD) measurements or moorings.

We start with a description of the elements of the glacial system model IGLOO 1.0, including the future and past forcings utilized in our paper (Section 2). In Section 3, we describe our initialization method, while Section 4 compares the simulated present-day surface elevation and velocity with observations. Further on, modelled basal properties are compared with findings of other works. In Section 5, we present projections of the GrIS sea level contribution, the GrIS total basal and surface runoff and the submarine melt rates for two GrIS outlet glaciers (Store and Helheim Glaciers). The paper closes with a discussion (Section 6) and the conclusions (Section 7).

## 2 Ice sheet model for Greenland including ocean and outlet glaciers (IGLOO), version 1.0

### 2.1 Overview of IGLOO

IGLOO is designed for better understanding the response of the Greenland glacial system to climate change on centennial timescales. It consists of sub-models for the Greenland ice sheet, the basal hydrology, the outlet glaciers and the turbulent meltwater plumes. For initialization via palaeo-runs, the model is forced by the temperature anomaly from GRIP ice core data, while for future projections we make use of data from climate models. The design of IGLOO is shown in Fig. 1.

While the ice sheet model SICOPOLIS is coupled bi-directionally with the model for basal hydrology HYDRO, the coupling between the turbulent meltwater plume and HYDRO is only implemented off-line yet. In the model set-up of this paper, SICOPOLIS does not simulate ice shelves. However, the impact of ocean temperature and subglacial ice discharge on submarine melt in future warming scenarios is investigated by the offline coupling with HYDRO for Store Glacier and Helheim Glacier (Fig. 2). The coupling between outlet glaciers and turbulent meltwater plumes, and future warming scenarios with this model configuration, are described in a accompanying paper by Beckmann et al. (2018a). The coupling between SICOPOLIS and outlet glaciers is not implemented yet.

In the following subsections, we will present the parts of IGLOO relevant for this paper, including coupling and forcing of these model components.

### 2.2 Ice sheet model SICOPOLIS version 3.3

SICOPOLIS (SImulation COde for POLythermal Ice Sheets; www.sicopolis.net) is a dynamic/thermodynamic ice sheet model that was originally created by Greve (1995, 1997) in a version for the GrIS. Since then, SICOPOLIS has been developed continuously and applied to problems of past, present and future glaciation of Greenland (e.g., Robinson et al., 2011), Antarctica

(e.g., Kusahara et al., 2015), the Eurasian ice sheet including subglacial water (Gudlaugsson et al., 2017), the entire Northern hemisphere (Ganopolski and Calov, 2011), the polar ice caps of the planet Mars and others (see www.sicopolis.net/publ for a comprehensive publication list). The description given here follows Greve et al. (2017) very closely.

The model simulates the large-scale dynamics and thermodynamics (ice extent, thickness, velocity, temperature, water content and age) of ice sheets three-dimensionally and as a function of time. It is based on the shallow ice approximation for grounded ice (Hutter, 1983; Morland, 1984) and the shallow shelf approximation for floating ice (Morland, 1987; MacAyeal, 1989). Recently, hybrid shallow-ice/shelfy-stream dynamics has been added as an option for ice streams (Bernales et al., 2017). The rheology is that of an incompressible, heat-conducting, power-law fluid (Glen's flow law; e.g., Greve and Blatter, 2009).

A particular feature of SICOPOLIS is its very detailed treatment of ice thermodynamics. A variety of different thermodynamics solvers are available, namely the polythermal two-layer method, two versions of the one-layer enthalpy method, the cold-ice method and the isothermal method (Greve and Blatter, 2016). The polythermal and enthalpy methods account in a physically adequate way for the possible co-existence of cold ice (with a temperature below the pressure-melting point) and temperate ice (with a temperature at the pressure-melting point) in the ice body, a condition that is referred to as "polythermal". It is hereby assumed that cold ice makes up the largest part of the ice volume, while temperate ice exists as thin layers overlying a temperate base. In the temperate ice layers, the water content is computed, and its reducing effect on the ice viscosity is taken into account (Lliboutry and Duval, 1985).

SICOPOLIS is coded in Fortran and uses finite difference discretization techniques on a staggered Arakawa C grid, the velocity components being taken between grid points (Arakawa and Lamb, 1977). For solving the thickness evolution equation, we added a further option to the SICOPOLIS code (Appendix A). The simulations of the GrIS discussed here are carried out in a stereographic plane (WGS 84 reference ellipsoid, standard parallel 71°N, central meridian 39°W), spanned by the Cartesian coordinates $x$ and $y$. The coordinate $z$ points upward.

## 2.3 Subglacial hydrology model HYDRO

HYDRO is a diagnostic model that determines the subglacial water fluxes instantaneously via the hydrological potential $\Phi$, which depends on the elevation potential and the water pressure $p_\mathrm{w}$ (Shreve, 1972):

$$\Phi = \rho_\mathrm{w} g\, b + p_\mathrm{w}, \tag{1}$$

with the ice base $b$, acceleration due to gravity $g$ and density of water $\rho_\mathrm{w} = 1000\,\mathrm{kg\,m^{-3}}$. The water pressure depends on the ice overburden pressure and the effective pressure $N$ (normal stress at the bed minus water pressure):

$$p_\mathrm{w} = \rho_\mathrm{i} g H - N, \tag{2}$$

wherein $\rho_\mathrm{i} = 910\,\mathrm{kg\,m^{-3}}$ is the density of ice and $H$ is the ice thickness.

Following previous authors such as Le Brocq et al. (2009) and Livingstone et al. (2013), we assume the water moving in a thin (a few mm) and distributed water film. Under this premise, the water pressure and the ice overburden pressure are in

equilibrium, and therefore the effective pressure is zero. This enables us to reformulate Eq. (1) as

$$\Phi = \rho_{\mathrm{w}} g\, b + \rho_{\mathrm{i}} g H, \tag{3}$$

and then computing the water flux with a simple flux routing scheme as described by Le Brocq et al. (2006). This approach is only valid at large (km) scales and is not able to include local processes such as channels.

The flux routing method requires that every cell has a defined flow direction and that, by successively following these directions, the boundary of the study area is reached. Therefore, local sinks and flat areas must be removed prior to applying the routing scheme. We accomplish this by using a Priority-Flood algorithm as described in Barnes et al. (2014), which fills depressions in a single pass and then add a small gradient to the resulting flats. Adding a gradient towards the outlet of the depression ensures that the hydraulic potential is altered in the smallest possible way. This procedure is a very efficient way to

guarantee that all water is drained into the ocean.

     The hydraulic potential is computed following Eq. (1), and we use the basal melt rates from SICOPOLIS as the water input for the routing scheme (see Section 2.5.1). The timescales of the water flow are much smaller than for the ice flow, thus, the steady-state water flux $\psi_{\mathrm{w}}$ can be obtained by integrating the basal melt rate along the hydraulic potential.

     From the resulting water flux $\psi_{\mathrm{w}}$, we can compute the water layer thickness $W$ (Weertman, 1972, 1966):

$$W = \left( \frac{12 \mu_{\mathrm{w}} \psi_{\mathrm{w}}}{\mathrm{grad}\, \Phi} \right)^{1/3}. \tag{4}$$

At locations where sinks in the hydraulic potential have been filled, we set $W$ to a very high value ($10\,\mathrm{m}$) to account for the presence of a subglacial lake.

## 2.4    Meltwater plume model

A further element of IGLOO is the line plume model by Beckmann et al. (2018b), based on Jenkins (2011). It simulates

the width-averaged submarine melt rate of a glacier and accounts for a uniformly distributed subglacial discharge along the grounding line. The plume model describes buoyancy-driven rise of subglacial meltwater until it reaches either neutral buoyancy or the water surface. Two counteracting processes control the maintenance or reduction of the plume buoyancy: submarine melting at the ice-ocean interface preserves the plume buoyancy, while simultaneously turbulent entrainment and mixing with the surrounding salty fjord water reduces it. The line plume equations are derived under the assumption that the plume is in

equilibrium and are thus time-independent. The melt rate is determined by the plume velocity and temperature, which adapts to the boundary conditions along the glacier front or under the floating tongue. As input parameters, it requires the submerged part of the glacier front $d$ and the subglacial discharge $Q$ that leaves the glaciers grounding line over the whole glacier width, and a temperature-salinity depth (TSD) profile close to the glacier. The determination of the input parameters of the plume model is described in section 2.5.2.

## 2.5 Coupling of model components

### 2.5.1 Coupling of SICOPOLIS with HYDRO

We use a slightly modified version of the Weertman-type sliding law proposed by Kleiner and Humbert (2014) to couple the basal hydrology model to the ice dynamics:

$$\boldsymbol{v}_{\mathrm{b}} = -f_{\mathrm{b}} C_{\mathrm{b}} |\tau_{\mathrm{b}}|^{p-1} \tau_{\mathrm{n}}^{-q} \tau_{\mathrm{b}}, \tag{5}$$

with the sliding velocity $\boldsymbol{v}_{\mathrm{b}}$, basal sliding parameter $C_{\mathrm{b}}$, basal shear stress $\tau_{\mathrm{b}}$, basal normal pressure $\tau_{\mathrm{n}}$ (assumed as the ice overburden pressure) and the stress and pressure exponents $p = 3$ and $q = 2$. We introduce the dimensionless factor

$$f_{\mathrm{b}} = f_T \left( (1 - c_{\mathrm{w}}) + c_{\mathrm{w}} f_{\mathrm{w}} \right), \quad c_{\mathrm{w}} \in [0, 0.9], \tag{6}$$

with

$$f_T = \exp\left(\frac{T - T_{\mathrm{pmp}}}{\nu}\right) \qquad \text{and} \qquad f_{\mathrm{w}} = \left(1 - \exp\left(-\frac{W}{W_0}\right)\right), \tag{7}$$

where $f_{\mathrm{T}}$ and $f_{\mathrm{w}}$ incorporate sub-melt sliding and basal hydrology respectively. Sub-melt sliding allows sliding below the pressure melting point $T_{\mathrm{pmp}}$ according to the decay parameter $\nu$ (Hindmarsh and Le Meur, 2001), whereas the basal hydrology term depends on the water layer thickness $W$ divided by a typical scale of the layer thickness $W_0$.

The parameter $c_{\mathrm{w}}$ in Eq. (6) is a weighting factor between "background sliding" – determined by $C_{\mathrm{b}}$ – and enhanced sliding due to the basal water layer. Using $c_{\mathrm{w}} = 0$ yields the standard model without any effect of basal hydrology, while $c_{\mathrm{w}} = 0.9$ leads to the same expression as Kleiner and Humbert (2014). In our simulation with basal hydrology, we apply their parameter value, i. e. we set $c_{\mathrm{w}} = 0.9$, while we specify the typical scale of the layer thickness by $W_0 = 0.005 \, \mathrm{m}$. Further, our decay parameter is $\nu = 1 °\mathrm{C}$.

The coupling is bi-directional. Basal melt $B$ (including the water drainage from the temperate basal layer of the ice sheet) computed by SICOPOLIS is used to calculate the thickness of the basal water layer in HYDRO, which in turn affects the basal sliding (Eq. 7). Components and data exchange of the complete coupled model IGLOO are illustrated in Fig. 1.

### 2.5.2 Off-line coupling of SICOPOLIS and HYDRO with the plume model

We establish a procedure of determining submarine melt rates with our line plume model (Section 2.4) for all Greenland outlet glaciers. This procedure applies only off-line yet, i. e., the input and output of the model components are exchanged manually via data files. To clarify, as this coupling is off-line, the sliding of ice (Section 2.5.1) is affected solely by basal melt, while the surface melt and basal melt can impact the meltwater plume.

For the subglacial discharge required by the plume model, we use HYDRO to route both the basal melt of SICOPOLIS and the surface runoff by MAR as basal water to the grounding lines of the outlet glaciers. We route on a monthly timescale to resolve seasonality. For the surface runoff, we assume that it penetrates directly down to the bedrock. Among others, Rignot

and Mouginot (2012) provide data of the geographical position of many outlet glaciers. We use these data to allocate the water leaving the ice sheet to the individual outlet glaciers.

Although we simulate future scenarios, the grounding line position is considered to be fixed for this procedure. Of course, for glaciers close to another that share the same catchment area, a moving grounding line position might have severe effects. We will account for these dynamic glacier processes in the next version of IGLOO.

## 2.6 Evaluating the data from the regional atmosphere model MAR

The ice sheet model needs the mean annual surface temperature and SMB as climate forcings at the surface. In addition, the plume model requires monthly runoff. Here, we explain how we derive these forcing fields and their gradients from data of simulations by the MAR regional climate model (Fettweis et al., 2013). These fields and their gradients serve to define our climate forcing of the GrIS for the past (Section 2.7) and for the future (Section 2.8).

Historical MAR simulations using different climate reanalysis products to define the boundary conditions for the regional simulations are available. The boundary conditions for MAR future projections up to 2100 are provided by the output of several CMIP5 general circulation models for different RCP scenarios. Since the MAR simulations are performed for fixed surface elevation of the GrIS, and we expect substantial changes in the ice elevation under future warming scenarios, we correct the regional model output for the change in surface elevation by applying the gradient method of Helsen et al. (2012). In their method, they derived a representative local elevation gradient of the SMB in each grid point from a regression of simulated SMB and surface elevation within a given radius. Helsen et al. (2012) did this separately for accumulation and ablation regimes. Here, we extend their method by applying it also to surface temperature and runoff.

In our scheme, the search radius is set to 100 km, but is extended until it includes at least 100 grid points if necessary. Our evaluation of the MAR data for the SMB revealed that the regression is in many cases not well defined for the accumulation regime. This issue can be also seen in Fig. 2 by Helsen et al. (2012). Therefore, we apply the gradient method only to the ablation regime and set the SMB elevation gradient for the accumulation regime to zero.

## 2.7 Past climate forcing and implied SMB of the GrIS

Our past climate forcing consists of the surface temperature and the SMB. By running the model over one glacial cycle, we determine an initial temperature-velocity field for our future warming scenarios. In particular, we determine the implied SMB for the present day, which is used in our future simulations as the climatological present-day SMB.

The surface temperature for the past simulation is computed from the sum of the climatological field of the present-day surface temperature simulated by MAR, the temperature anomaly from the GRIP ice core and our temperature elevation correction obtained from the present-day MAR simulations:

$$T_s(x,y,t) = T_{s\ \text{MAR(rean)}}^{\text{Clim } 1961-1990}(x,y) + \Delta T_{\text{GRIP}}(t) + \left(\frac{\partial T_s}{\partial z}\right)_{\text{MAR(rean)}}^{\text{Clim } 1961-1990}(x,y)\ \Delta z(x,y,t). \tag{8}$$

The elevation correction in the last term of Eq. (8) is the surface temperature elevation gradient (Section 2.6) from the MAR reanalysis data times a surface elevation difference, which reads

$$\Delta z(x,y,t) = z(x,y,t) - z_0(x,y), \tag{9}$$

with the surface elevation $z$, simulated with the ice sheet model SICOPOLIS, and the observed surface elevation $z_0$. For the observed surface elevation, we use the one by Bamber et al. (2013), which is the same as that utilized by Fettweis et al. (2013).

Here, the SMB $M$ is defined by relaxing the ice sheet's surface elevation towards the observed surface elevation as

$$M(x,y,t) = \frac{z_0(x,y) - z(x,y,t)}{\tau_{\mathrm{relax}}}, \tag{10}$$

where $\tau_{\mathrm{relax}}$ is a relaxation constant. With this relaxation method, we follow Aschwanden et al. (2013, 2016). Outside the ice sheet, we assign the high negative value of $M = -1000 \,\mathrm{m\,ice/yr}$, which prevents the ice to flow outside its domain. Running the ice sheet model in time by applying Eq. (10) specifies an iteration for the surface elevation and the SMB. Therefore, we do not need any further input here.

Applying the forcing fields for the surface temperature (Eq. 8) and the SMB (Eq. 10), we run the model over one glacial cycle. As we start the palaeo-simulation with the observed surface elevation, the simulated surface elevation relaxes soon towards the observed one. The relaxation constant $\tau_{\mathrm{relax}}$ determines how close the simulated surface elevation is to the observed one. When the model reaches its present-day state ($t = 0$), we obtain the present-day implied SMB $M_{\mathrm{impl}}$, which is used in the future simulations, as

$$M_{\mathrm{impl}}(x,y) := M(x,y,0). \tag{11}$$

This SMB field corresponds approximately to the observed SMB, but compensates for errors of the ice sheet model. We will discuss further the relaxation approach and its limitations in Sections 3 and 6.

Outputs of this procedure are the present-day implied SMB and a nearly present-day topography set (surface and bedrock elevation) belonging to this implied SMB. Later on, the present-day implied SMB field (Eq. 11) is added to the anomaly forcing of future climate simulations (see Eq. 13).

## 2.8 Future climate forcing of the GrIS

The surface temperature forcing is computed from the climatological temperature of MAR simulations for 1961–1990 forced by the ERA reanalysis boundary conditions, the anomalies from MAR simulations forced by CMIP5 model output and a temperature elevation correction as:

$$
\begin{aligned}
T_s(x,y,t) = T_{s\,\mathrm{MAR(rean)}}^{\mathrm{Clim\,1961-1990}}(x,y) \quad &+\quad (T_{s\,\mathrm{MAR(CMIP5)}}(x,y,t) - T_{s\,\mathrm{MAR(CMIP5)}}^{\mathrm{Clim\,1961-1990}}(x,y)) \\
&+\quad \left(\frac{\partial T_s}{\partial z}\right)_{\mathrm{MAR(CMIP5)}}(x,y,t)\;\Delta z(x,y,t).
\end{aligned}
\tag{12}
$$

Here, the temperature elevation correction is determined via the product of the surface temperature elevation gradient (Section 2.6) of the MAR model with boundary condition from the CMIP5 models and the elevation anomalies simulated with the ice sheet model SICOPOLIS. As for the palaeoclimate, $\Delta z(x,y,t)$ are the simulated surface elevation anomalies (Eq. 9).

The SMB for future projections is computed as the sum of the implied SMB, simulated SMB anomalies relative to the reference period 1961–1990 and an elevation SMB correction as follows:

$$M(x,y,t) = M_{\text{impl}}(x,y) \quad + \quad (M_{\text{MAR(CMIP5)}}(x,y,t) - M_{\text{MAR(CMIP5)}}^{\text{Clim 1961}-1990}(x,y))$$
$$+ \quad \left(\frac{\partial M}{\partial z}\right)_{\text{MAR(CMIP5)}} (x,y,t) \quad \Delta z(x,y,t). \tag{13}$$

Similar to temperature, the elevation SMB correction is calculated from the SMB elevation gradient (Section 2.6) of the MAR model with boundary condition from the CMIP5 models, multiplied by the simulated surface elevation anomalies.

Surface runoff is computed for each month from the climatological runoff of MAR simulations for 1961–1990 forced by the ERA reanalysis boundary conditions, the anomalies from MAR simulations forced by CMIP5 models output and a runoff elevation correction (Section 2.6), which again is computed similarly to the temperature elevation correction:

$$R(x,y,t) = R_{\text{MAR(rean)}}^{\text{Clim 1961}-1990}(x,y) \quad + \quad (R_{\text{MAR(CMIP5)}}(x,y,t) - R_{\text{MAR(CMIP5)}}^{\text{Clim 1961}-1990}(x,y))$$
$$+ \quad \left(\frac{\partial R}{\partial z}\right)_{\text{MAR(CMIP5)}} (x,y,t) \quad \Delta z(x,y,t). \tag{14}$$

Negative runoff values, which can result from this approach, are set to zero.

Figure 3 shows time series derived from the MAR data. During the 20th century, all curves show rather minor changes in average, besides a visible climate variability. This is in line with general knowledge (e .g. Box et al., 2009; Box and Colgan,
2013). In the 21st century, the anomaly of the SMB over Greenland is strongest for CanESM2, weakest for NorESM1, and MIROC5 lies in between. Of course, these 21st-century warming trends correspond to IPCC AR5 (Collins et al., 2013) because the MAR forcing is from the CMIP5 models. The annual average temperature change over Greenland is stronger than the global one.

Over the years 1900–1949, MAR provides data only for MIROC5, and after the year 2100, MAR does not provide data for
any of the GCMs used. To obtain complete forcings for the years 1900–2300, we closed the data gaps with an extrapolation procedure (described in detail in Appendix B). We extended our forcings to the year 2300 for the sake of comparability with other studies involving ice sheet models, in which also long projections were made (e. g. Edwards et al., 2014).

### 2.9  Future climate forcing of the plume model

As future forcing of the plume model, we employ the subglacial discharge from HYDRO and SICOPOLIS (Section 2.5.2) under
the RCP 8.5 scenario (Section 2.8) from MAR with MIROC5 only. Additionally, a scenario of the temperature and salinity profiles is needed to project future submarine melting. Even for the present day, measurements inside fjords are rare and do not cover all of Greenland's fjords. We use CTD profiles close to the glaciers obtained for the year 2016 for Store Glacier (data from NASA's OMG mission(https://omg.jpl.nasa.gov/portal/)) and the year 2012 for Helheim Glacier (Carroll et al., 2016). For the ocean warming scenario, we assume a linear temperature trend of 0.03 °C per year over the years 2000–2100 for the
entire profiles.

The 3 °C ocean warming in 100 years lies in the upper range found by Yin et al. (2011) for SE and W Greenland. The determined temperature and salinity profiles, in combination with the HYDRO output, serve as the input parameters for the line plume model to determine present and future submarine melting for the Greenland outlet glaciers.

## 3   Model initialization via palaeo-runs

For the initialization of the ice sheet model, we use the forcings for the surface temperature and the SMB as described in Section 2.7. Using a standard setting of SICOPOLIS for palaeo runs, isostatic depression and rebound of the lithosphere due to changing ice load is modelled assuming a local lithosphere with relaxing asthenosphere with an isostatic time lag (LLRA approach, Le Meur and Huybrechts, 1996). For the geothermal heat, we use the spatial dependent data by Purucker (2012). In order to cover one full glacial cycle, we run the model over 135 kyrs. Initial conditions of these runs are the present-day

ice thickness and elevation by Bamber et al. (2013). The original data with 1 km horizontal resolution are downsampled to 5 km and 10 km grid spacings. To perform a simulation in 5 km horizontal resolution over the entire glacial cycle with the hybrid model is illusive, as it takes 1 day for 100 model years on one HLRS2015 Lenovo NeXtScale nx360M5 processor. Therefore, we opt to perform the first 130 kyrs of the glacial cycle in 10 km horizontal resolution with the classical shallow ice approximation (SIA) employing the diffusivity method with an over-implicit ice-thickness solver. The last 5 kyr of the

palaeo-run are performed in 5 km horizontal resolution. As we use different model hierarchies and settings, we devote some more explanation to these last 5 kyr.

  During the last 5 kyrs of the run, we have three switches: one for refining the horizontal resolution, one for switching from SIA mode to hybrid mode, and a further one for switching from relaxing ice surface to free ice surface. The first switch at 5 kyr BP refines the horizontal resolution of the model from 10 km to 5 km. The second switch at 500 years BP changes from SIA

to hybrid mode and additionally applies the mass conservation scheme for the evolution equation of ice thickness (Eq. A1). The third switch, which releases the relaxing ice surface to free development, is imposed at 100 years BP (year 1900). We introduced the three switches at different times because they represent different regime changes.

  We chose the time of the resolution switch furthest back in time (5 kyr BP) to allow the ice temperature to adapt to the refined basal topography. Indeed, by comparing the simulated present-day basal temperature in 10 km resolution with that in

5 km resolution from our simulation with the switches, one can observe that the 5 km basal temperature shows a much finer structure of the temperate basal regions at locations of outlet glaciers and their catchment areas; see also Fig. S1c and d in Calov et al. (2018). As these temperate regions determine areas with basal sliding, enabling fast flow important for ice dynamics, we regard a time of 5 kyr as sufficient for resolving the major characteristics of the ice temperature field in the 5 km resolution.

  Another important aspect for the choice of an adequate timing of the switches is the rate of sea level change, which quantifies

the model drift and shows directly how the model recovers from the transition shock. We demonstrate this by inspecting different switching times and comparing our favourite initialization run with a test simulation (Fig. 4). While our initialization run has the switch from SIA to hybrid at the year 1500 and the switch from relaxing surface to free surface at the year 1900, the test simulation has both of these switches at the year 1900. Both curves of the rate of sea level change first rise fast to a

maximum and then drop slowly towards smaller values. For the test simulation, the maximum is larger (0.06 cm/yr) compared to that of our initialization run (0.05 cm/yr). More importantly, the test simulation has not enough time to recover. While the rate of sea level change of our initialization run at the year 2000 amounts only to $-0.0007$ cm/yr, it is much larger (0.03 cm/yr) for the test simulation. In future projections, a drift of 0.03 cm/yr would yield a non-negligible contribution to modelled sea level rise of 3 cm in 100 yrs. Therefore, our initialization with separate switches for regime changes from SIA to hybrid and from relaxed to free surface is much more favourable and thus used for our projections (Section 5.1).

The choice of the relaxation constant rests on numerous simulations in 10 km horizontal resolution in SIA mode, running the model over one glacial cycle until the present day. Figure 5 shows the root mean square error (RMSE) in surface elevation and the total difference in SMB (the total implied SMB over the GrIS minus the total SMB simulated by MAR). With increasing relaxation constant, the RMSE in surface elevation increases moderately, while the total difference in SMB decreases strongly, i. e., there is a tradeoff between the RMSE in elevation and the total difference in SMB.

Figure 6 shows the spatial differences between the observed and modelled surface elevation and SMB for different relaxation constants. Again, the tradeoff for representing both surface elevation and SMB is visible. While the simulated elevation is very close to the observation for small relaxation constants, the SMB deviation is very high, even in the interior of the ice sheet, where the deviations reach the amount of magnitude of the accumulation rate. Therefore, too small relaxation constants should be excluded. For larger relaxation constants, both difference fields become smoother, but rather high deviations in surface elevation appear over vast areas of the GrIS. Therefore, too high relaxation constants should be excluded too. The spatial differences in Fig. 6b, e are from the medium relaxation constant of $\tau_{\mathrm{relax}} = 100$ years. One can see that the simulated surface elevation is too low over the north-western and south-eastern marginal regions (Fig. 6b). Over these regions, the simulated velocities are too high, which is compensated by an increased implied SMB (Fig. 6e) compared to the SMB from MAR. For most regions of the Northeast Greenland Ice Stream (NEGIS), the opposite situation occurs. The simulated surface elevation is too high, while the implied SMB is mostly negative over that region, which compensates the too low simulated velocities, see also Fig. 7. Again, the choice of the relaxation constant $\tau_{\mathrm{relax}}$ is a tradeoff between the errors in the surface elevation and the errors in the implied SMB.

## 4 Present-day Greenland ice sheet

Here, we present our optimal simulation of the GrIS using the SICOPOLIS model version 3.3 with hybrid dynamics and the model for basal hydrology (HYDRO). Both models are fully coupled (see Section 2.5.1), and the horizontal resolution is always 5 km from now on. In the hybrid mode, a threshold of $r_{\mathrm{thr}} = 0$ applies to the slip ratio (Eq. 8 in Bernales et al. (2017)), i. e., the SStA equations are solved over the entire ice sheet, and the ice velocities are the weighted sum from the SIA and SStA velocities with the slip ratio as weight. The boundary conditions and initialization method to yield the present-day GrIS are described in Sections 2.7 and 3, respectively. As relaxation constant for the surface elevation we use $\tau_{\mathrm{relax}} = 100$ years. Optimal values for the sliding parameters are found by minimizing the error of simulated horizontal surface velocities for

values $> 50$ m/yr, using observations by Rignot and Mouginot (2012). For such velocities, we expect basal sliding and hybrid ice dynamics to be relevant. We found $C_b = 25$ m/(Pa yr) to be optimal for the hybrid model with basal hydrology.

By design of the initialization, the simulated surface elevation compares overall well with the observed one, see Fig. 7a,b. However, as our surface relaxation method leaves the ice sheet's surface a certain degree of freedom (see also Fig. 6), the simulated ice surface over Summit and South Dome as well as on the ridge in between them is slightly lower. The simulated surface velocities ($< 2$ m/yr) over the slow flow regions in the vicinity of the ridges are somewhat smaller compared to the observed surface velocities. Such (small) mismatches also appear with other higher-order models, even in higher resolution (Aschwanden et al., 2016). As we adjusted the sliding parameter $C_b$ to match velocities higher than 50 m/yr with observations, it is obvious that the model cannot resolve every detail over the slow flow regions. Still, the model resolves the major flow patterns of the GrIS, including the flow over the catchment area of the outlet glaciers and the fast flow of the major outlet glaciers and ice streams. Only the smaller-scale outlet glaciers, e. g. in north-west Greenland, are not fully resolved. Since we have excluded ice shelves in SICOPOLIS, we cannot reproduce outlet glaciers with floating tongues, such as Petermann, Nioghalvfjerdsbræ and Zachariæ Isstrøm. The NEGIS is the only larger scale feature which we cannot reproduce properly. This feature cannot be simulated without additional assumptions (see the Discussion section).

Figure 8 zooms in Jakobshavn Isbræ and the two major outlet glaciers Helheim and Kangerdlugssuaq. Here, the ability of the model to resolve the catchment areas of these outlet glaciers in a 50 to 500 m/yrs range can be seen in more detail. However, the high-velocity patterns near the glacier termini do not fully match the simulations. In particular, the tributaries of Helheim and Kangerdlugssuaq glaciers and the tip of Jakobshavn Isbræ appear rather smooth compared to the observation.

Fast flow mainly appears over regions with a temperate ice bed. The simulated basal temperature in Fig. 9a shows a pattern which agrees basically with the reconstruction by MacGregor et al. (2016). Regions where there is basal melt, mainly caused by basal friction, exhibit a 1 to 5 mm thick water layer (Fig. 9b). There is a pronounced thickening of the water layer with our Shreve-flow modelling toward major ice streams and outlet glaciers, which is most visible for NEGIS, Jakobshavn Isbræ and Helheim Glacier. Moreover, smaller outlet glaciers like Store Glacier and Daugaard-Jensen Glacier receive intensified basal water supply too. The red dots over central-east Greenland correspond to sinks in the hydrological potential. These sinks are interpreted as subglacial lakes, following Livingstone et al. (2013) who reported similar results. In our simulations, these sinks can appear over a frozen base too, because we operate HYDRO with an option that allows computation of the hydrological potential over the entire ice area. This has the advantage that all basal water can safely reach the ocean. Allowing a water layer over a frozen bed for such sinks in the hydrological potential affects ice dynamics only very little because fully developed sliding appears only over temperate basal areas, while sub-melt sliding decays rapidly with decreasing temperature (Section 2.5.1).

## 5  Greenland glacial system projections

### 5.1  Projections of the GrIS's sea level contribution

For our projections of the contribution of the GrIS to global sea level rise, the GrIS is forced by SMB anomalies and surface temperatures derived from the MAR regional climate model (Section 2.8), making use of the initial ice sheet configurations explained in Section 3. As for the last 500 years of initialization, the fully coupled hybrid model including basal hydrology is utilized. Outside of the present-day GrIS area, similarly to the initialization, the prohibiting negative SMB is applied. In Fig. 10, we show the GrIS sea level contribution referenced to the year 2000. The control simulation forced solely with the implied SMB illustrates the characteristics of our initialization method. The sea level change from the control run is always small compared to that of all projections, see Fig. 10. Only compared to the RCP 4.5 projection over 300 years, a small change in ice volume is visible for the control run (Fig. 10b). At the year 2300, the control run shows a sea level rise of 3.0 mm. This corresponds to a model drift of 1.0 mm per hundred years between 2000 and 2300. Despite such a small change, we correct our simulated sea level contribution of the GrIS in the simulation with MAR forcing for the implied-SMB-only simulations. This correction is based on the ice volumes from the future simulations forced with the MAR data and the ice volume from the run forced with implied SMB only. All simulated ice volumes – those from the runs with MAR and that from the run forced with implied SMB only – are referenced to their respective year 2000 volumes.

Our projections of the GrIS sea level contribution for the year 2100 are close to simulations with a fixed present-day GrIS applying the cumulative SMB method (Church et al., 2013). This is in line with simulations with an active ice sheet model by Goelzer et al. (2013), who found that SMB is the major factor determining the GrIS sea level contribution over the 21st century. Our simulated GrIS sea level contribution for 2100 ranges from 1.9 cm (RCP 4.5, NorESM1) to 13.0 cm (RCP 8.5, CanESM2), see Table 2. Still, the ice dynamics (deformation and sliding velocities) plays a role in our simulations, indirectly via the SMB change. This can be seen when comparing the simulations with and without elevation SMB correction $(\partial M/\partial z)\Delta z$, Eq. (13). Ignoring the elevation SMB correction diminishes simulated 21st-century GrIS sea level contribution between 0.4 and 1.7 cm. Of course, this effect is strongest for the extreme RCP 8.5 scenario together with CanESM2, which is the CMIP5 model used here with the largest SMB anomaly. Interestingly, the relative effect of the elevation SMB feedback at the year 2100 is smaller for RCP 8.5 compared to RCP 4.5. This corresponds to the findings by Vizcaino et al. (2015), who used the ECHAM5 AOGCM coupled to an ice sheet model for their projections.

At the end of the 23rd century, the contribution of the GrIS to sea level rise ranges from 3.5 cm to 76.4 cm. The importance of the elevation SMB feedback clearly increases with the elapsed time of the projections, as the respective curves with $\partial M/\partial z$ on/off diverge more and more from each other. For RCP 8.5 with CanESM2, the relative increase of additional loss in ice volume due to elevation SMB correction nearly triples from 2100 to 2300, from 15 % to 40 %. This increase of the relative effect of this feedback with projection time was also observed by Edwards et al. (2014) and Vizcaino et al. (2015). Detailed numbers for the sea level contributions of the GrIS for the years 2100 and 2300 are listed in Table 2.

Overall, our simulations show a strong dependence of the GrIS sea level contribution both on the RCP scenarios and on the models used to force MAR.

## 5.2 Projections of the GrIS's total basal and surface runoff

For these projections, we use the basal melt from the two simulations by SICOPOLIS (Section 5.1) forced by the MAR data for which MAR used the MIROC5 GCM under the RCP 8.5 scenario. Surface and basal melt are routed over the ice base and distributed to the GrIS outlet glaciers. The details are explained in Section 2.5.2. Figure 11 depicts the total subglacial discharge split into surface runoff and basal melt. Over the entire simulation time, the total basal melt is small and ranges between 10 and 12 Gt/yr, while the total surface runoff increases up to a peak value of 1700 Gt/yr. Note that, after the year 2100, the total surface runoff decreases due to the shrinking ice sheet area. Simultaneously, the effect of the elevation SMB feedback becomes more important after the year 2100. At 2100, the difference between the total surface runoffs with and without elevation SMB feedback amounts to only 150 Gt/yr, while the same difference for 2300 nearly doubles to 290 Gt/yr.

## 5.3 Projections of submarine melt rate for the GrIS outlet glaciers Helheim and Store

Here, we inspect the impact of global warming under the RCP 8.5 scenario for two outlet glaciers: Helheim Glacier and Store Glacier. In detail, we analyse the impact of both subglacial discharge and ocean warming – as single and combined effects – on the submarine melt rate of these outlet glaciers. While the subglacial discharge comes from simulations with SICOPOLIS and HYDRO under the RCP 8.5 scenario, the ocean warming originates from a scenario similar to RCP 8.5 (Section 2.9). For analysing the impact of the elevation SMB feedback on submarine melt, the plume model is forced by subglacial discharge computed with and without the surface elevation correction of surface runoff (Eq. 14). We calculate all submarine melt rates under the assumptions of both glaciers being tidewater glaciers (no floating tongues) and of their grounding-line depths and widths remaining constant in time. These depths and widths are acquired from present-day observations and amount to 500 m depth and 5 km width for Store Glacier (Chauché et al., 2014) and 650 m depth (Carroll et al., 2016) and 6 km width (Straneo et al., 2016) for Helheim Glacier. We chose the entrainment parameter to be $E_0 = 0.036$ as recommended by Beckmann et al. (2018b).

Figures 12 and 13 illustrate the monthly subglacial discharge and the temperature profiles for the years 2000 and 2100 and the resulting submarine melt rates for the RCP 8.5 scenario. For both glaciers, the increasing subglacial discharge and the increasing ocean temperature have an about equal effect on the rising submarine melt, with the ocean temperature becoming more important towards the end of the year 2100. However, the combined effect of increased subglacial discharge and temperature exceeds the single effects alone. As a result, submarine melt exhibits a 2.5-fold increase for Helheim Glacier and a 4-fold increase for Store Glacier in the year 2100 (Figs. 12c and 13c). Although for the year 2000 the amount of basal melt (38 $\mathrm{m}^3/\mathrm{s}$ for Helheim, 5 $\mathrm{m}^3/\mathrm{s}$ for Store) is small compared to summer subglacial discharge (818 $\mathrm{m}^3/\mathrm{s}$ for Helheim, 439 $\mathrm{m}^3/\mathrm{s}$ for Store), it has a significant effect on the annual submarine melt rate. Due to the basal melt in the winter months (including early spring and late autumn), the submarine melt rate enlarges in those months substantially as illustrated by Fig. 14 for Helheim Glacier. The slight increase in subglacial discharge for all months (Fig. 14a) shows clearly the biggest increase in submarine melt rate for the winter months (Fig. 14b) due to the cubic root dependence of submarine melt rate on subglacial discharge (Jenkins, 2011). On the annual average, this effect leads, for the year 2000, to an increase of submarine melt for Helheim

Glacier by 40 % and for Store Glacier by 20 % compared to the case when basal melt was not accounted for (Figs. 12c and 13c). At 2100, the elevation SMB feedback increases the submarine melt rate of Helheim Glacier and Store Glacier from 787 by 76 to 863 m/yr and from 804 by 81 to 723 m/yr, respectively (Figs. 12c and 13c). With about 10 % for both outlet glaciers, this effect is relatively small. However, as Fig. 11 suggests, the effect will become more important for the submarine melt rate after the year 2100.

In these experiments, the future submarine melt rate was calculated assuming a constant glacier terminus position and geometry. These calculation have to be seen as a first approximation because we neglect several factors that may influence the submarine melt rate. For instance, if the glacier retreats, the resulting grounding line depth may change depending on the underlying bedrock. Another factor that might change the melt rate estimation considerably is the distribution of subglacial discharge within the year. Here, we assumed no time lag in between runoff and its emergence as subglacial discharge. Due to the cubic root dependence of submarine melting on subglacial discharge, we already see the possible strong effect of basal runoff from the ice sheet on the distribution of the submarine melt rate of an outlet glacier over the year (see Fig. 14). Thus, an inefficient drainage system that is delayed by, e. g., storage of water in subglacial lakes (Nienow et al., 2017) might affect the seasonal distribution of subglacial discharge and thus the annual submarine melt rate substantially.

## 6   Discussion

In Section 3, we investigated the role of the relaxation constant for initialization. For very small relaxation constants, i. e., an essentially fixed ice surface, the difference between implied and observed SMB at present day becomes very large (more than 2000 Gt/yr, compared to an insignificant amount for $\tau_{\text{relax}} = 100$ years). Note that the present-day magnitude of observed total SMB is only about 500 Gt/yr (e .g. Ettema et al., 2009). This means that computations with fully fixed surface should be treated with care, as the required total implied SMB is very high. A further advantage of the relaxation of the ice surface is that this smoothes the ice surface because it has a certain degree of freedom due to the relaxation constant while solving the ice thickness equation. A similar smoothing effect when running an ice sheet model with free surface evolution over 100 yrs was already observed by Calov and Hutter (1996). Furthermore, they demonstrated that a smooth ice surface avoids irregular variations in the vertical velocity field.

For our initialization method, we made a number of simplifications. We ignored changes in surface elevation and spatial extent of the GrIS during the past glacial cycle. Anyhow, the elevation changes during the last glacial cycle over central parts of Greenland like Summit (about 100 m, Raynaud et al., 1997) or Dye 3 (some 100 m, Vinther et al., 2009) were rather small compared to the ice thickness. These rather small differences in elevation limit the inaccuracies in the ice temperature field caused by our relaxation approach, also because the maxima of these changes appeared about 10,000 years ago during the Holocene. Furthermore, due to slowly flowing ice there, a slightly different temperature-velocity field will not play a too important role for the overall ice dynamics. Also, ignoring the larger areal extension (Funder and Hansen, 1996) at the Last Glacial Maximum (LGM) should not affect the ice temperature field too strongly because over the more peripheral regions of the ice sheet the velocities are rather high and the LGM is 21,000 years ago. Here, the ice temperature will adapt faster and

has more time to adapt compared to the central regions of the ice sheet. The assumption that the derived present-day elevation correction is valid over the entire glacial cycle is reasonable. We have examined this by evaluating the temporal dependence of the vertical temperature gradient from the MAR data. Indeed, for the future, this temperature gradient does not change too strongly in time, which does not mean that the elevation correction itself can be neglected here. Anyhow, for the palaeo-run,

our ice surface is rather constant in time due to our relaxation approach, which makes the elevation correction small in this case. As a further simplification, we applied the temperature anomaly from the GRIP record to the entire GrIS. This is a standard approach, but still, we are aware that this anomaly is spatially dependent in reality. However, possible errors in this anomaly will become less and less important the nearer the palaeo-simulation approaches the present day. That our present-day temperature-velocity field is reasonable is supported by the fact that it lies in the range found by MacGregor et al. (2016).

In this context, we would like to repeat that our initialization serves the purpose to have the simulated surface elevation at present-day as close as possible to the observed one to minimize the drift in the future projections.

In our simulations, we cannot reproduce the NEGIS correctly. Certainly, one reason is that we do not optimize the surface velocity by a spatially dependent basal sliding coefficient. With spatially dependent basal sliding coefficients, other studies such as Price et al. (2011) and more recently Peano et al. (2017) simulated the NEGIS in better agreement with observations. Nowadays, there are process-oriented approaches to capture effects important for basal sliding. For example, stronger basal melting at the onset of the NEGIS caused by increased geothermal heat due to a palaeo-hotspot (Rogozhina et al., 2016) could be one factor speeding up the simulated NEGIS velocity.

For our 300-year sea level projections, which reach beyond the 21st century, we prolong the forcing data of the MAR model until the year 2300. Because we merely held the forcing constant between 2101 and 2300, the real RCP 8.5 forcing could be larger, i.e., we expect our simulations with the RCP 8.5 scenario to be a lower estimate of sea level contribution of the GrIS, i.e., the estimate is a rather conservative one. Most certainly, even all our projections including RCP 4.5 are a conservative estimate because a full coupling with ice–ocean interactions is missing in our model yet, and Fürst et al. (2015) found that ocean warming caused additional mass loss of the GrIS in his projections applying a parameterization of ocean warming.

Our additional sea level rise for the year 2100 due to elevation SMB feedback is somewhat higher than that by Le clec'h et al. (2017), who used the regional model MAR actively coupled to an ice sheet model for their simulations. For the year 2100, Edwards et al. (2014) give an additional contribution to sea level rise of the GrIS due to this feedback ranging between 0.25 and 0.32 cm under the SRES A1B scenario; we excluded their outlier of 0.1 cm. Still, their estimates of this feedback are rather small compared to our ones lying between 0.6 and 1.7 cm, which were produced with the RCP 8.5 scenario though. The SRES A1B scenario is somewhat more moderate that the RCP 8.5 scenario. However, this cannot explain such low numbers for the elevation SMB feedback. As demonstrated to be important by Le clec'h et al. (2017) with fully interactive two-way coupling, this feedback deserves a detailed inspection in the future.

Our presented projections for the GrIS contribution to global sea level rise in the 21st century (1.9-13.0 cm) are consistent with previous publications. However, they do not account for the dynamic response of Greenland outlet glaciers to ocean warming and increase of subglacial discharge. This effect will be account for in a forthcoming paper. We also intend to couple the 3-D ice sheet model SICOPOLIS with the 1-D model for many outlet glaciers.

# 7 Conclusions

We introduced the coupled Greenland glacial system model IGLOO 1.0 designed to describe the most important parts of the Greenland glacial system: the ice sheet, the subglacial hydrological system, the outlet glaciers and the ice-ocean interaction in the Greenland fjords. The applicability of the hybrid mode of the ice sheet model SICOPOLIS 3.3 to the Greenland ice
sheet was demonstrated. Full coupling between the ice sheet model and the model of subglacial water HYDRO has been accomplished, while the coupling between HYDRO and the meltwater plume works only off-line yet.

As initialization, we used a relaxation method similar to Aschwanden et al. (2013), but with a somewhat higher relaxation constant of 100 years. For this choice of the relaxation constant, we varied it systematically and investigated the resulting model behaviour by inspecting the RMS error in surface elevation as well as the difference between total simulated SMB and
total SMB from the MAR regional climate model. It showed that, for a relaxation constant of 100 years, the deviation of our simulated total SMB from the MAR SMB is about zero, while – at the same time – the RMS of the simulated error in surface elevation stays reasonably small. Additionally, we showed that medium-value relaxation times lead to smooth 2-D fields of the implied SMB, while for too small relaxation times the fields become rather noisy, and for too large relaxation times regional deviations of the simulated elevation from the observed one become relatively large (RMS error of 95 m for $\tau_{\mathrm{relax}} = 300$ years,
see Fig. 5a).

Furthermore, we performed projections of the contribution of the GrIS to sea level rise until the year 2300 with hybrid ice dynamics forced by SMB anomalies from the MAR regional model. For the RCP 4.5 and 8.5 scenarios generated by MAR, three CMIP5 GCMs with different climate sensitivity were applied. Altogether, our projected GrIS sea level contribution for the year 2100 obtained with elevation SMB feedback ranges from 1.9 to 13.0 cm, and for the year 2300 from 3.5 to 76.4 cm.
The effect of elevation SMB feedback contributes clearly to our simulated contribution of the GrIS to sea level rise. Generally, its impact increases in the long run with decreasing surface elevation (see Table 2).

Moreover, we demonstrated the importance of the different factors determining the increase of the melt rate of Greenland outlet glaciers under the extreme RCP 8.5 scenario, using Store and Helheim Glaciers as examples. It showed that the knowledge of near-terminus temperature and subglacial discharge in the fjord are both about equally important to determine the
future melt of these two outlet glaciers. This underlines the importance of our approach with the Greenland system model IGLOO 1.0.

*Code and data availability.* SICOPOLIS is available as free and open-source software at www.sicopolis.net. The HYDRO module is not included in the repository yet. The MAR data used as the basis for our forcing are available at ftp://ftp.climato.be/fettweis/MARv3.5/Greenland/.

## Appendix A: Mass-conserving scheme for ice thickness evolution

We included a new numerical scheme into SICOPOLIS 3.3, which discretizes the advection term of the ice thickness equation by a strictly mass-conserving scheme in an upwind flux form:

$$A = \frac{(\bar{v}_x(i+1/2,j)H_x^+ - \bar{v}_x(i-1/2,j)H_x^-)\Delta y + (\bar{v}_y(i,j+1/2)H_y^+ - \bar{v}_y(i,j-1/2)H_y^-)\Delta x}{\Delta x \Delta y}, \tag{A1}$$

where $A$ is the advection term and $\bar{v}_x$, $\bar{v}_y$ are the $x$- and $y$-components of the depth averaged velocity, respectively. Further, $\Delta x$ and $\Delta y$ are the horizontal spacings. The upwind coefficients read:

$$H_x^- = \begin{cases} H(i-1,j), & \bar{v}_x(i-1/2,j) \geq 0, \\ H(i,j), & \bar{v}_x(i-1/2,j) < 0, \end{cases} \qquad H_x^+ = \begin{cases} H(i,j), & \bar{v}_x(i+1/2,j) \geq 0, \\ H(i+1,j), & \bar{v}_x(i+1/2,j) < 0, \end{cases}$$

$$H_y^- = \begin{cases} H(i,j-1), & \bar{v}_y(i,j-1/2) \geq 0, \\ H(i,j), & \bar{v}_y(i,j-1/2) < 0, \end{cases} \qquad H_y^+ = \begin{cases} H(i,j), & \bar{v}_y(i,j+1/2) \geq 0, \\ H(i,j+1), & \bar{v}_y(i,j+1/2) < 0, \end{cases}$$

with the ice thickness $H$. The pairs $(i,j)$, $(i+1/2,j)$ etc. indicate the indices of the staggered Arakawa C grid.

## Appendix B: Filling the data gaps of the MAR forcing for initialization and future simulations

While the MIROC5 model provides data starting at the year 1900, the CanESM2 and NorESM1 models start later in time at the year 1950. Therefore, there are no CanESM2 and NorESM1 data for the years 1900–1949. The MAR data consist of the MAR fields, which are longitudinally-latitudinally distributed fields for annual mean surface temperature, SMB and monthly surface runoff. Because the climate over Greenland changed relatively little during the 20th century, we took the years 1950–1999 as sampling interval and determined randomly years out of this interval. The MAR fields of a single CMIP5 model for these random years are assigned to MAR fields of the subsequent years inside the target interval, which is defined by the years 1900–1949. By this, we close the data gaps of CanESM2 and NorESM1 for the years 1900–1949.

For the years 2101–2300, there are no direct scenario data available from MAR for any of the three used CMIP5 models. For the MAR fields from the RCP 4.5 scenario, we apply the same random procedure as described above to fill the data gaps, but with the sampling interval 2091–2100 and the target interval 2101–2300. While for RCP 4.5 forcing the climate-warming trend over the 21st century is moderate, it is stronger for RCP 8.5, see Fig. 3.

We found that the variability for the years 2100–2300 was relatively high for RCP 8.5 scenarios, see Fig. A1, panel a. Therefore, we modified our random procedure for the MAR fields from RCP 8.5. For RCP 8.5 forcing, we took only those MAR fields from the sampling interval that belong to the lower 70 % of their total SMB anomaly. In other words, we excluded the MAR fields for the upper 30 % of the total SMB anomaly, see Fig. A1, panel b. Mathematically this reads

$$\Delta M_{\text{tot}} > \Delta M_{\text{tot}}^{\text{ave}} + 0.3 \cdot (\Delta M_{\text{tot}}^{\text{max}} - \Delta M_{\text{tot}}^{\text{min}})/2, \tag{B1}$$

with the temporal average, the maximum and the minimum of the anomaly of the total SMB, $\Delta M_{\text{tot}}^{\text{ave}}$, $\Delta M_{\text{tot}}^{\text{min}}$ and $\Delta M_{\text{tot}}^{\text{max}}$, respectively.

*Competing interests.* The authors declare no competing interests.

*Acknowledgements.* We wish to thank Fiamma Straneo and Dustin Carroll for providing temperature and salinity profiles of Helheim glacier, Xavier Fettweis for hosting the website with the MAR data, and Ciarion Linstead for technical support with library installations. Further, we are grateful for helpful suggestions from the scientific editor Carlos Martin and two anonymous reviewers. J. B., R. C. and S. B. were funded by the Leibniz Association grant SAW-2014-PIK-1. M. R. was supported by the Helmholtz Alliance Climate Initiative (REKLIM). M. W. was supported by the Bundesministerium für Bildung und Forschung (BMBF) grants PalMod-2.1 and PalMod-2.2. R. C. was funded by the Bundesministerium für Bildung und Forschung (BMBF) grants PalMod-1.1 and PalMod-1.3. R. G. was supported by Japan Society for the Promotion of Science (JSPS) KAKENHI grant numbers 16H02224 and 17H06104, and by the Arctic Challenge for Sustainability (ArCS) project of the Japanese Ministry of Education, Culture, Sports, Science and Technology (MEXT).

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

## Tables

**Table 1.** Abbreviations in Fig. 1.

| Abbreviations | Physical meaning |
|---|---|
| $z_0$ | Observed present-day elevation of the GrIS |
| $z$ | Simulated elevation of the GrIS |
| $\Delta T_{\mathrm{GRIP}}$ | Reconstruction of temperature anomaly from GRIP ice core |
| $\Delta T_{\mathrm{s}}$ | Anomaly of surface temperature simulated by MAR |
| $\Delta M$ | Anomaly of SMB simulated by MAR |
| $\Delta R$ | Anomaly of runoff simulated by MAR |
| $T_{\mathrm{s}}$ | Surface temperature |
| $M$ | Surface mass balance (SMB) |
| $R$ | Surface runoff |
| $Q$ | Subglacial discharge into the given fjord |
| $B$ | Bottom melt simulated by SICOPOLIS |
| $W$ | Thickness of basal water layer |
| $T$ | Ocean temperature (function of depth) |
| $S$ | Ocean salinity (function of depth) |
| $d$ | Submerged part of the outlet glaciers |
| $M_{\mathrm{s}}$ | Submarine melt of the outlet glaciers |

**Table 2.** Simulated GrIS contribution to sea level rise for the years 2100 and 2300 in cm. Rows specify the different GCMs used by MAR. Columns list the RCP scenarios used by the MAR GCMs and whether we excluded or included the elevation SMB feedback $\partial M/\partial z$ in our simulation.

| MAR GCM | Year 2100 [cm] | | | | Year 2300 [cm] | | | |
|---|---|---|---|---|---|---|---|---|
| | RCP 4.5 | | RCP 8.5 | | RCP 4.5 | | RCP 8.5 | |
| $\partial M/\partial z$ | off | on | off | on | off | on | off | on |
| NorESM1 | 1.5 | 1.9 | 4.0 | 4.6 | 1.8 | 3.5 | 18.8 | 25.5 |
| MIROC5 | 3.7 | 4.3 | 7.7 | 8.8 | 8.5 | 10.8 | 33.7 | 46.3 |
| CanESM2 | 4.6 | 5.6 | 11.3 | 13.0 | 11.2 | 17.1 | 54.6 | 76.4 |

**Figures**

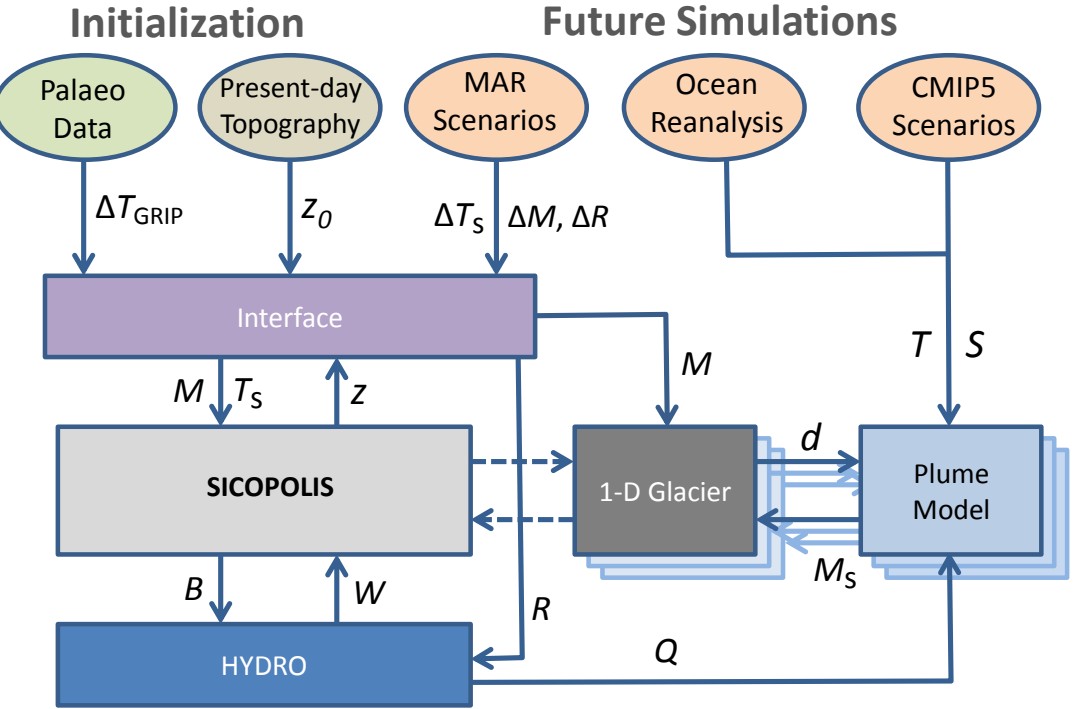

**Figure 1.** Flow diagram of the model IGLOO and the interaction between its components. The 1-D outlet glacier and plume models are generic models, i. e., they can be applied to each outlet glacier of the Greenland ice sheet. Coupling between the ice sheet model and the generic outlet glacier models is not implemented yet, denoted by dashed arrows. In this paper, coupling between HYDRO and the plume model is off-line. Simulations with the coupled generic outlet glacier models and plume models as well as details on the coupling between them are described in Beckmann et al. (2018a). The exchange variables are explained in Table 1.

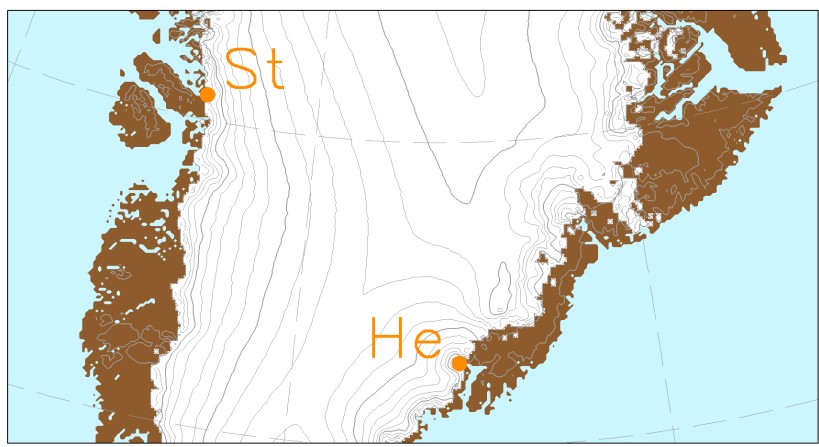

**Figure 2.** Geographical position of the outlet glaciers mentioned in the main text. "St" indicates the location of Store Glacier, while "He" marks the position of Helheim Glacier.

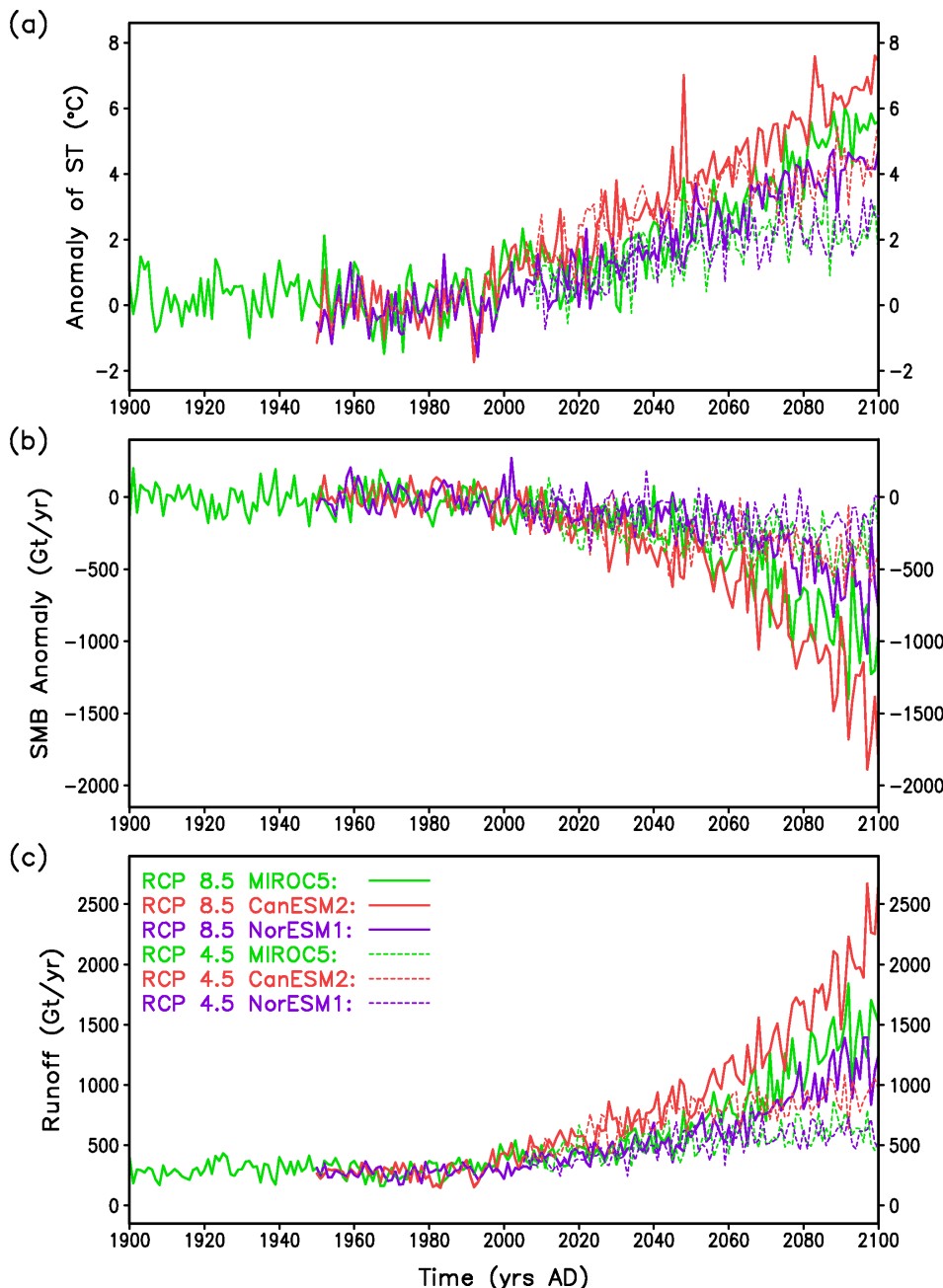

**Figure 3.** Forcings derived from the MAR regional model. (a) Anomaly of annual average surface temperature, (b) total annual SMB anomaly, and (c) total annual runoff. Anomalies are taken with respect to the period 1961–1990 from the respective CMIP5 models. RCP 8.5 scenarios are indicated by the solid lines, while RCP 4.5 scenarios are shown by the dashed lines.

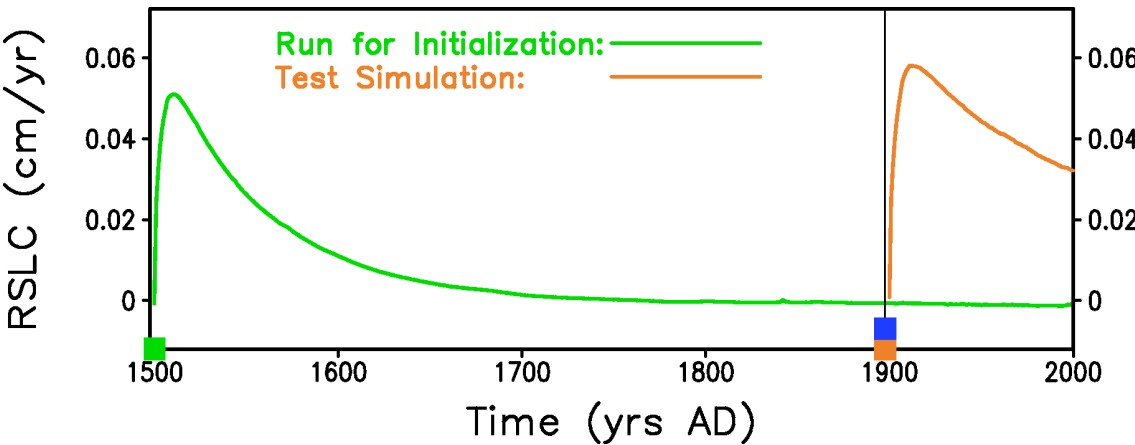

**Figure 4.** Time series of the rate of sea level change produced by the GrIS, illustrating the impact of the timing of the last two switches during the initialization. The green curve is from the simulation used for initialization of our future projections, which switches from the shallow ice approximation to the hybrid mode at the year 1500 (green square). The orange curve is from a test simulation (not used for initialization of our future projections), where we switch from the shallow ice approximation to the hybrid mode at the year 1900 (orange square). Both simulations switch from relaxing surface to free surface at the year 1900 (blue square), i. e., for the test simulation, the two switches appear at the same time.

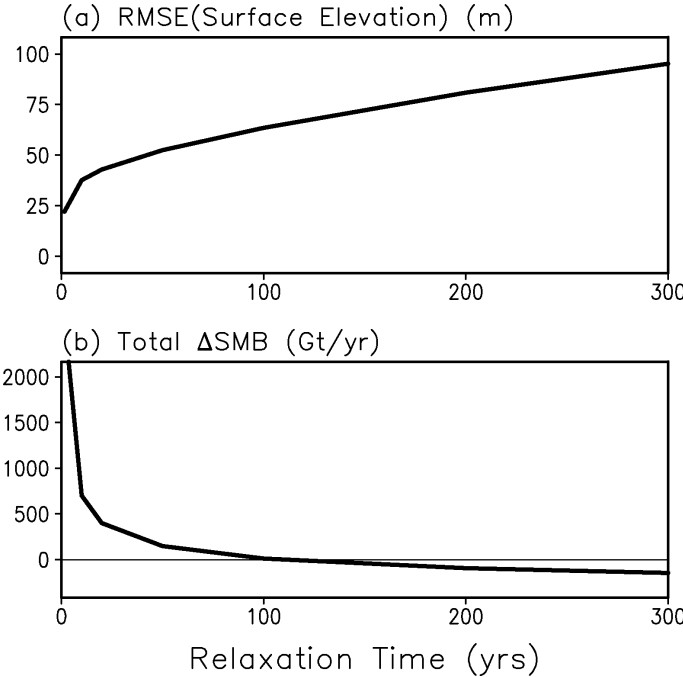

**Figure 5.** Total ice sheet quantities at present day against relaxation constant. (a) Root mean square error (RMSE) of modelled to observed surface elevation. (b) Total difference between our simulated surface mass balance and the SMB from the regional model MAR using ERA reanalysis 1961-1990 climatology.

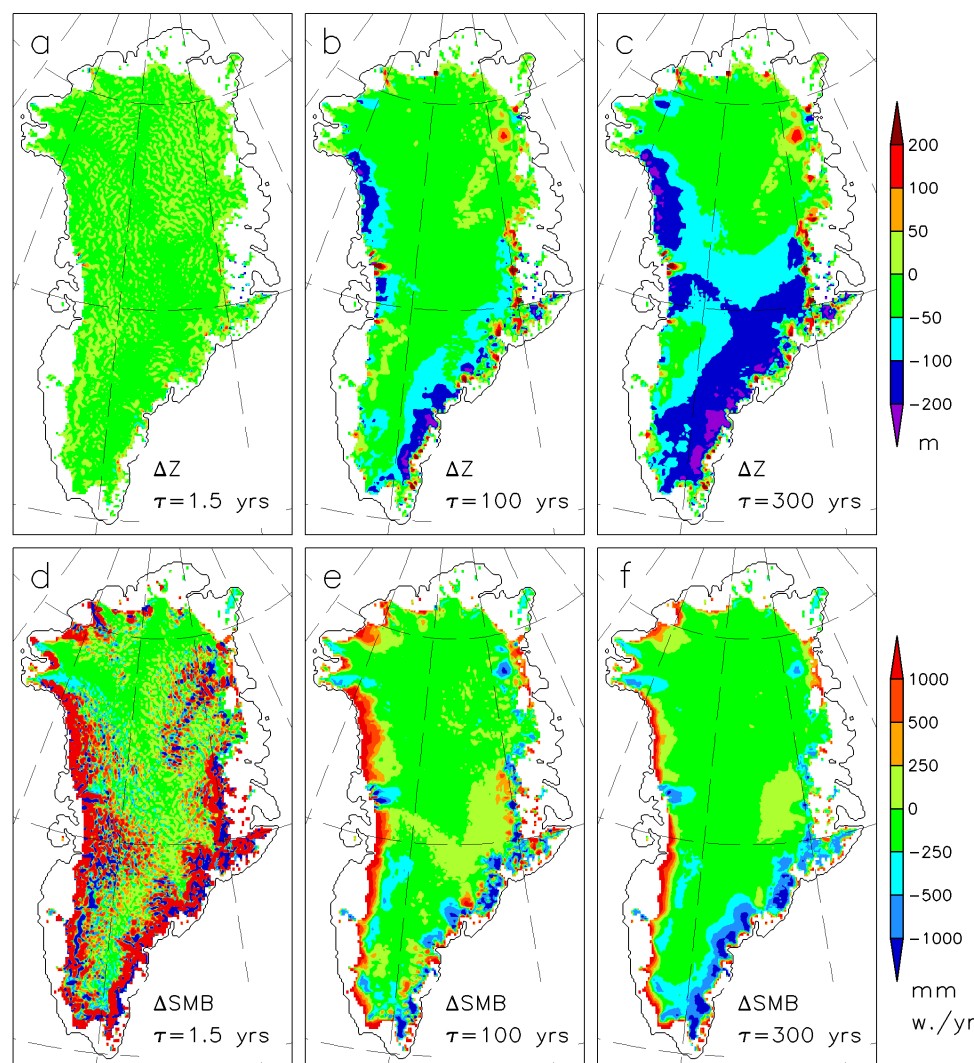

**Figure 6.** Differences between simulated and observed present-day 2-D fields for various relaxation constants, i. e., 1.5, 100 and 300 years. (a), (b) and (c): deviation of surface elevation from observed. (d), (e) and (f): deviation of our implied SMB from the SMB from the regional model MAR.

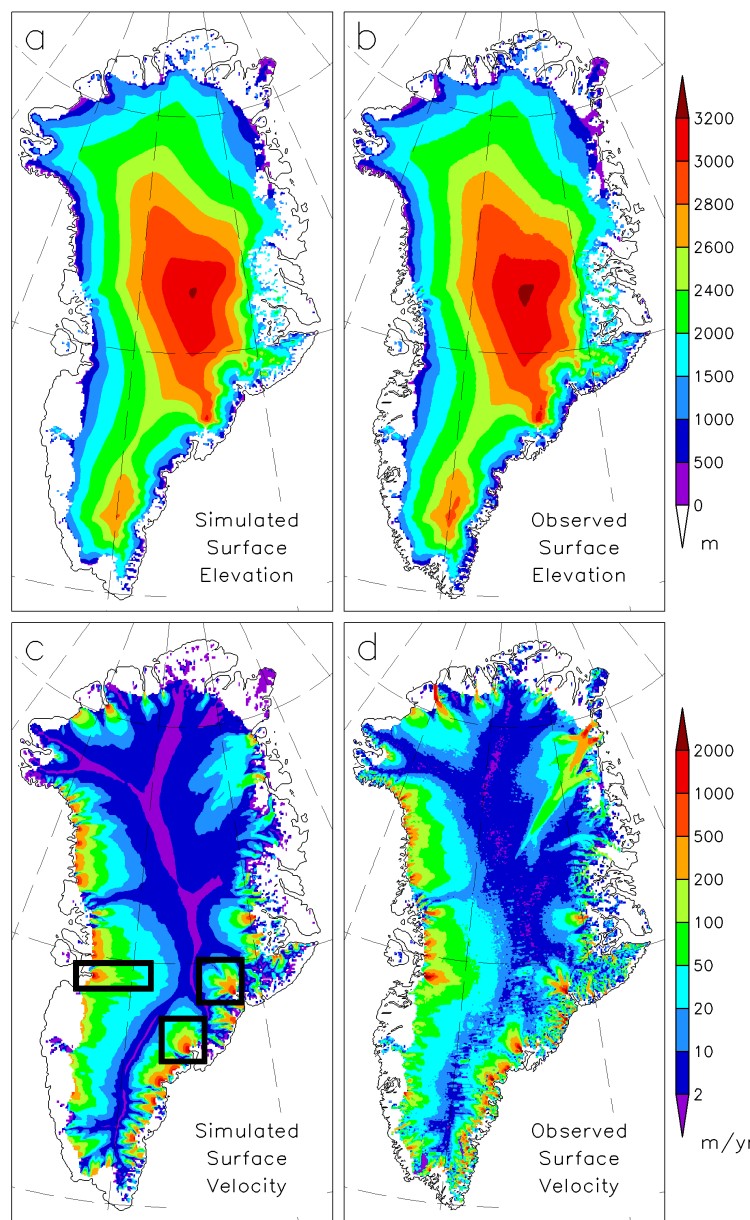

**Figure 7.** Comparison of simulated and observed present-day 2-D fields with 100 yrs relaxation constant. (a) Simulated surface elevation, (b) surface elevation by Bamber et al. (2013), (c) simulated horizontal surface velocity, and (d) horizontal surface velocity by Rignot and Mouginot (2012). The rectangles in panel c indicate the regions around Jakobshavn Isbræ, Helheim Glacier and Kangerdlugssuaq Glacier, which are enlarged in Fig. 8.

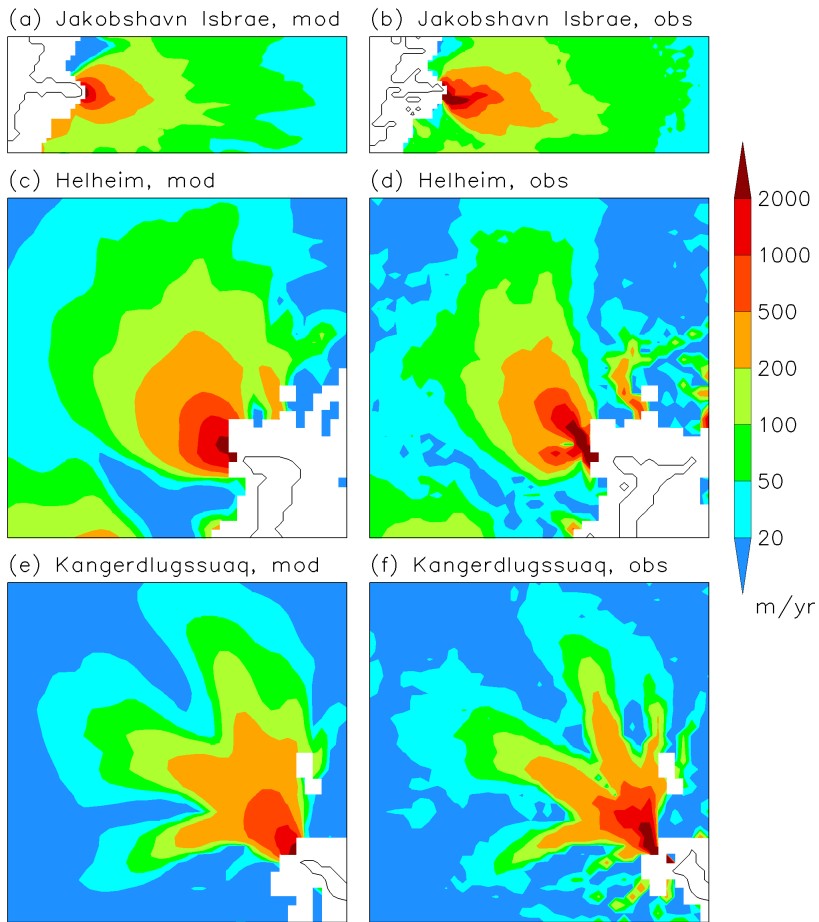

**Figure 8.** Comparison of simulated and observed (Rignot and Mouginot, 2012) velocity for major ice streams and outlet glaciers. Left side: modelled, right side: observed. (a, b) Jakobshavn Isbræ, (c, d) Helheim Glacier, and (e, f) Kangerdlugssuaq Glacier. The location of the three regions in Greenland is shown in Fig. 7c.

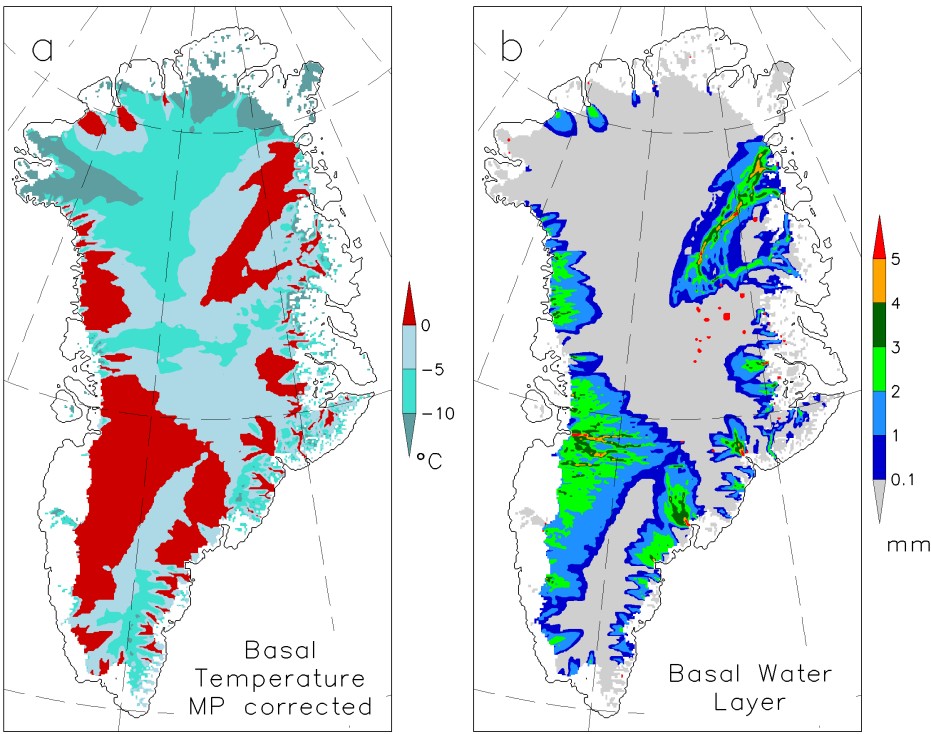

**Figure 9.** Simulated 2-D basal fields. (a) basal temperature relative to pressure melting (in °C), (b) thickness of basal water layer (in mm).

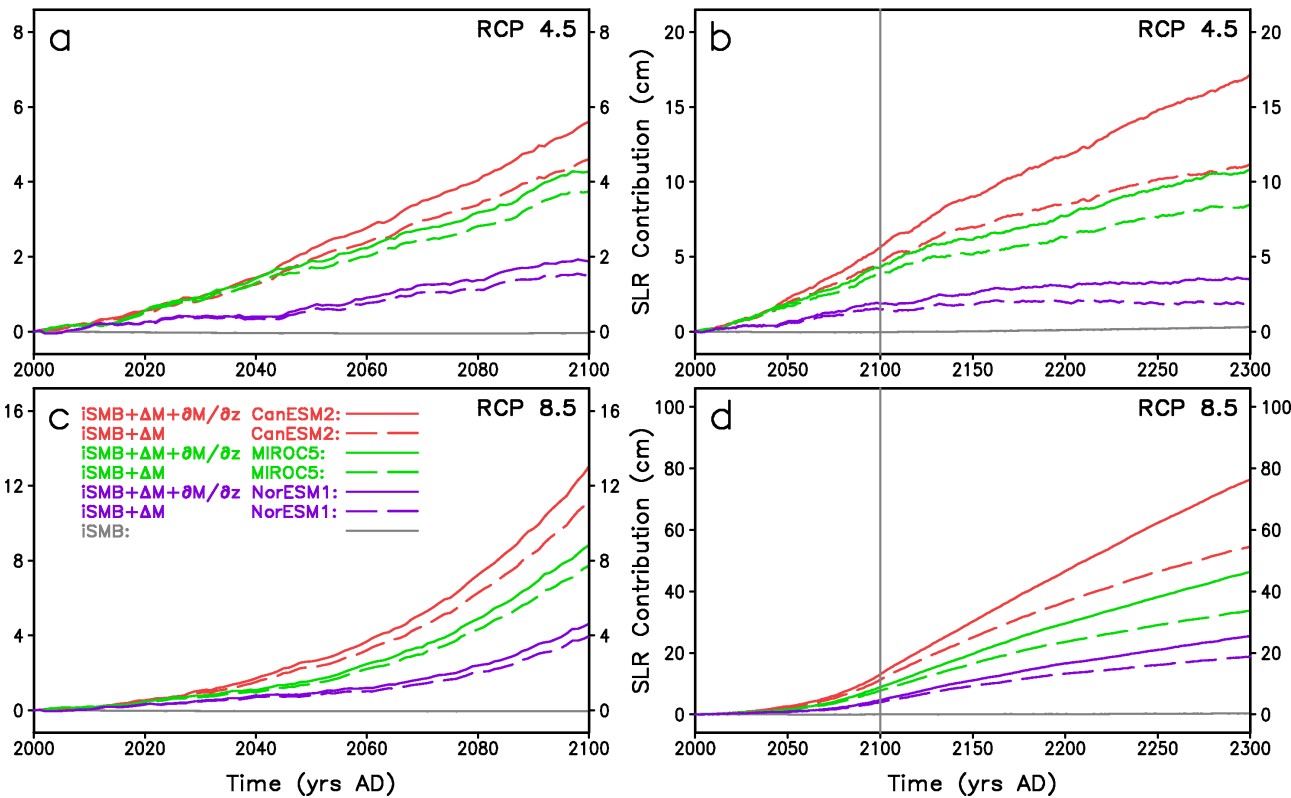

**Figure 10.** Contribution of the Greenland ice sheet to future sea level rise under MAR forcing for different scenarios. Sea level rise is referenced to the year 2000. Beyond 2100, the forcings of the projections are from prolongations of the original MAR data (see main text for details). This is indicated by the vertical grey line at the year 2100 in panels (b) and (d). RCP 4.5 projections: (a) years 2000–2100 and (b) years 2000–2300. RCP 8.5 projections: (c) years 2000–2100 and (d) years 2000–2300. The different CMIP5 general circulation models utilized by MAR are indicated by colours. Different line characteristics specify optimal simulations with (solid) and without (long dashed) elevation correction for the SMB. The grey curves in panels (a) to (d) indicate a control simulation with solely the implied SMB (iSMB) as forcing. All simulations are with hybrid ice dynamics and HYDRO basal hydrology.

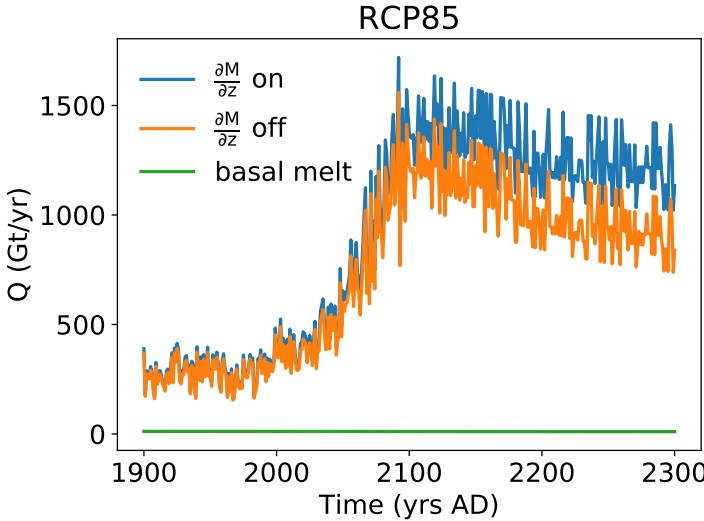

**Figure 11.** Time series of the components of subglacial discharge. The total basal melt (green) is nearly constant in time and ranges from 10 to 12 Gt/yr. Total surface runoff with surface elevation SMB feedback (blue) and without the feedback (orange).

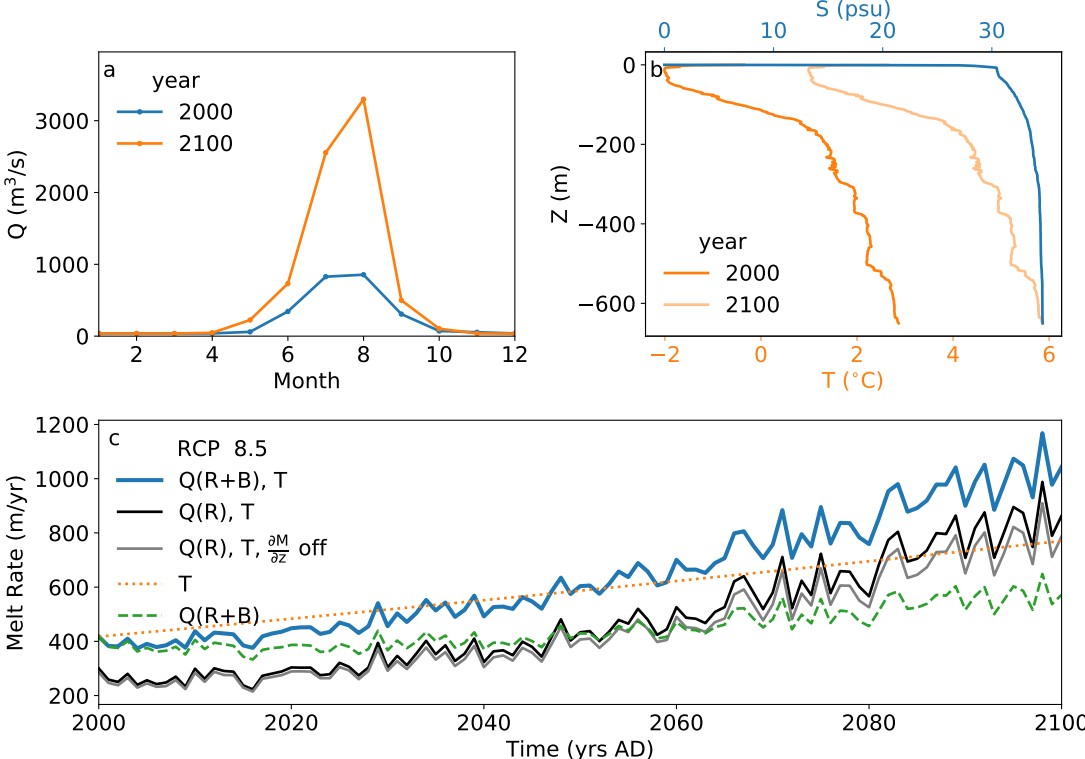

**Figure 12.** (a) Monthly subglacial discharge derived from runoff and basal melt (R+B) for Helheim Glacier and the scenario RCP 8.5 in the years 2000 and 2100. (b) Temperature-depth and salinity-depth profiles obtained from measurements for the years 2000 and 2100. The corresponding submarine melt rates are depicted in (c). The effects of increased temperature and discharge only (orange dotted and green dashed lines respectively), as well as the combined effect (solid lines) are displayed until the year 2100. Melt rates with subglacial discharge from solely surface runoff are depicted in black. Melt rates of subglacial discharge containing only surface runoff that was calculated without the surface elevation feedback are depicted in grey.

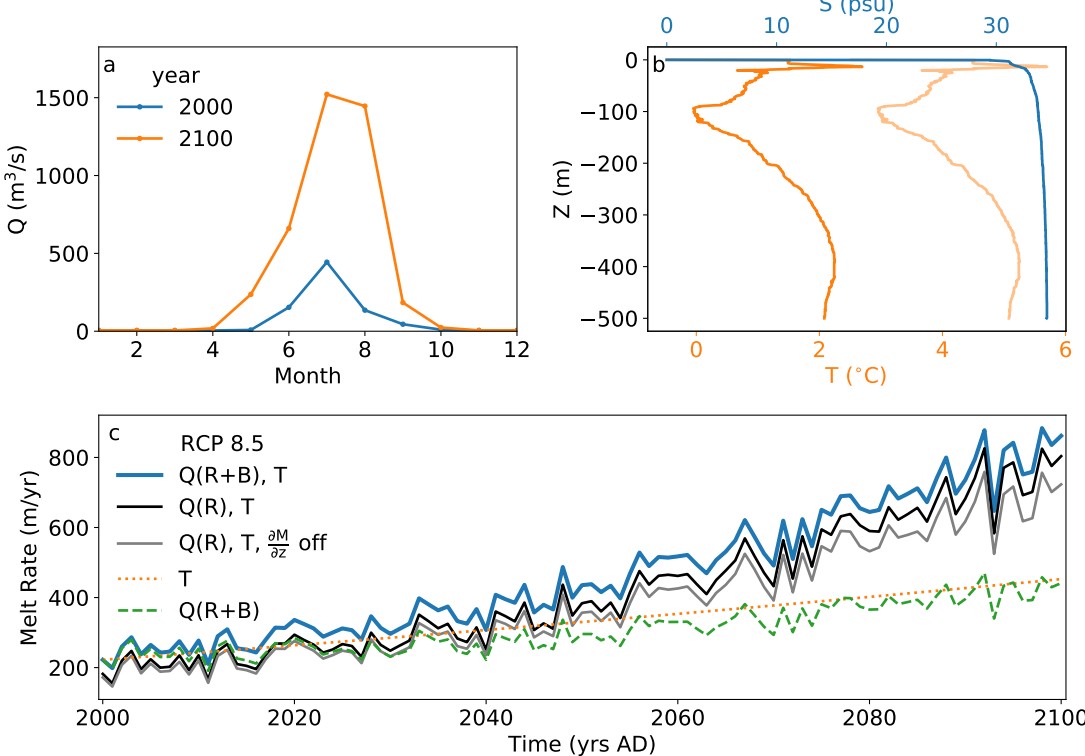

**Figure 13.** Similar to Figure 12, but for Store Glacier.

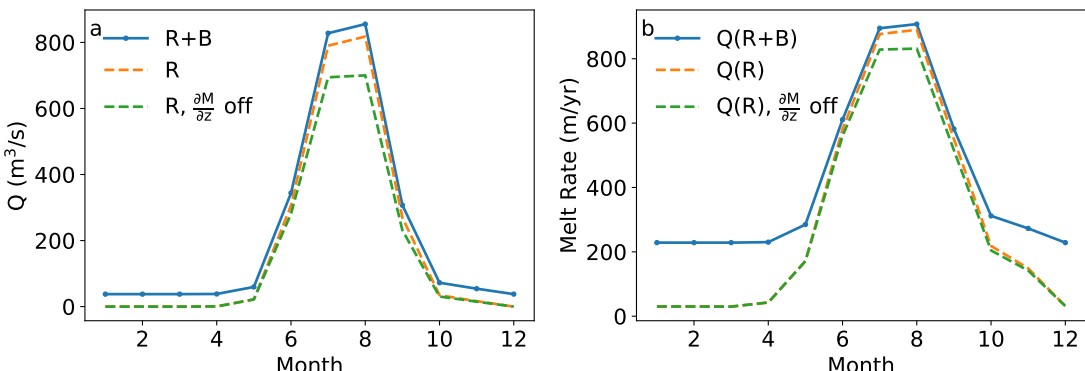

**Figure 14.** Subglacial discharge of Helheim Glacier (a) for the year 2000 determined by runoff (R) only (dashed lines), with and without surface elevation feedback (orange, green) and runoff together with basal melt (R+B, blue solid line). The corresponding submarine melt rates (b) with the same line colour and line style.

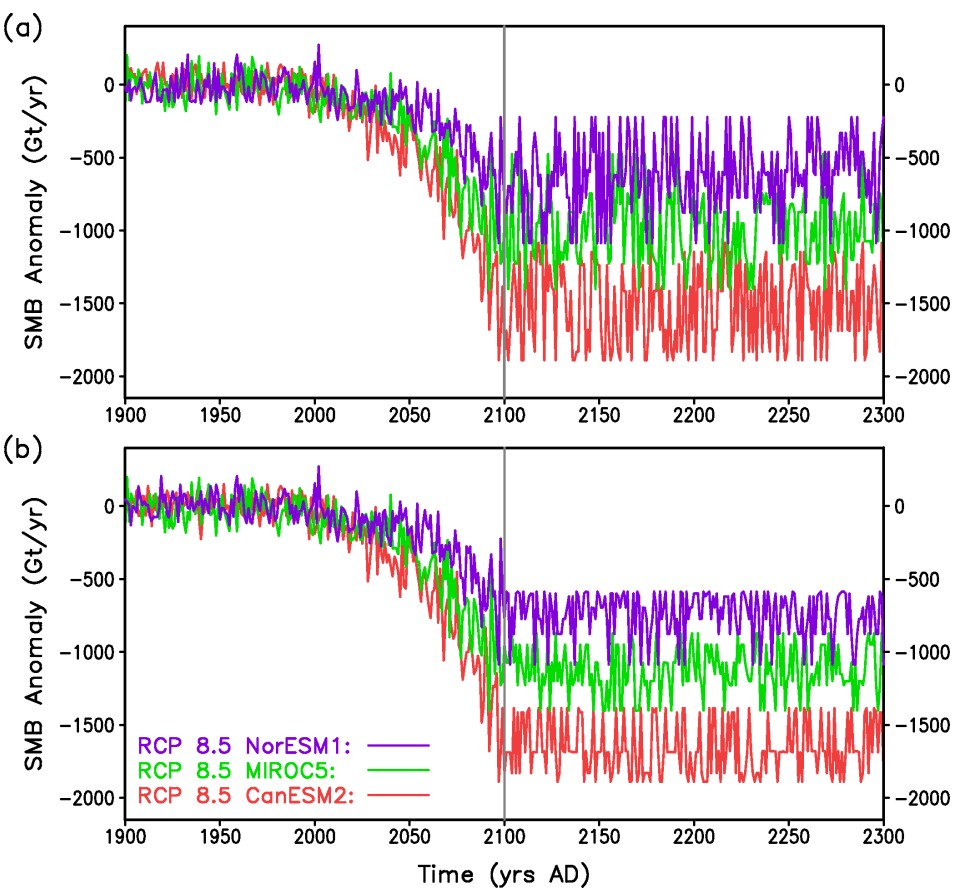

**Figure A1.** Prolongation of the MAR forcing illustrated for the SMB anomaly. (a) For the years 2101–2300, the SMB anomalies are taken from random years in between 2091–2100. (b) Variability for the years 2101–2300 reduced by 30 %. The vertical grey line at the year 2100 indicates the beginning of the prolongation.