# Peer review of "Simulation of the future sea level contribution of Greenland with a new glacial system model"

_The Cryosphere, 2018_

## Referee Comment (RC1) · Anonymous Referee #1 · 3 May 2018

General comments

This manuscript describes a model study with a new glacier system model that consists of polythermal ice sheet model SICOPOLOS version 3.3 coupled to a model of basal hydrology, HYDRO, and a parameterization of submarine melt for marine-terminating outlet glaciers. The system model is forced with climatology from MAR, 1961-1990, forced with ERA-reanalysis boundary conditions and future projections are made with surface mass balance anomalies derived from RCP4.5 and 8.5 scenarios created by the regional climate model MAR forced with boundary conditions from three CMIP5 models. Projections for sea level rise contribution are made until 2300 and total sub-glacial discharge analysed as well as for two outlet glaciers, Helheim and Store where the offline coupled model is applied as case study area. The authors state that the

outlet glacier model will be applied to further outlets in the future. This topic is timely and it is very important to develop models that are capable of including feedback mechanisms that can have significant effect on the future evolution of the large ice sheets as well as the climate system (here the ocean), in order to make realistic projections of future evolution of the Greenland ice sheet.

Specific comments

The manuscript is well organized and clearly written, but I find the discussion of the results and comparison with data or other studies quite qualitative and think it would improve the paper if the authors would make more quantitative analysis of their results, see numerous comments below.

The description of the forcing scheme is not clear (section 2.6) as it is not clear how the model applies the temperature defined in equation 8, as it seems that according to equation 10 the mass balance is only the difference between the modelled and measured elevation, divided by relaxation constant. Is this really the case, or is missing a description of a positive degree day method to compute the surface mass balance during the spin up period and current equation 10 would be one term of that forcing?

The naming of the implied flux (equation 11) is confusing, suggest to call it something that indicates surface mass balance.

The introduction section is comprehensive and gives a good overview of the current state of development of large scale ice sheet models, and there is an impressive reference list for this study. I find, however, that the first part of the introduction should have more references for the general statements (page 1, lines 21 and 22, as well as page 2 lines 1 and 2) or at least indicate that these are not the only papers stating those broad things, with "e.g." before that one reference.

Minor comments: Page 1 line 19 suggest to replace "melting" with "retreat"

Page 2 line 4-5, suggest to add something about the peripheral glaciers, that are not

attached to the ice sheets, as those are actually contributing considerably to the sea level rise, see for example: Machguth, H., P. Rastner, T. Bolch, N. Mölg, L. Sandberg Sørensen, G. AÃřalgeirsdottir, J. H. van Angelen, M. R. van den Broeke and X. Fettweis. 2013. The future sea-level rise contribution of Greenland's glaciers and ice caps. Environ. Res. Lett. 8, 025005 doi:10.1088/1748-9326/8/2/025005

Page 2, line 6, suggest to add "a" before interplay

Page 2, line 8 suggest to replace "thought" with "intended"

Page 2, line 10, suggest to replace "treating" with "treat"

Page 2, line 11 suggest to replace "fast" with something like "computationally effiecent"

Page 2 line 5, suggest to add references after "studies exist"

Page 2, line 19, suggest to replace "can serve" with "serves"

Page 2 line 20 suggest to delete "as" after "and"

Page 2 line 22 suggest to replace "elevation" with "ice thickness"

Page 2 line 23 suggest to replace "elevation" with "ice thickness"

Page 2 line 24 suggest to replace "due" with "according"

Page 2 line 25 suggest to add "the" before "projected"

Page 2 line 27, suggest to delete "in" after "yielding" and add "of" before "the present-day"

Page 2, line 28 suggest to replace "avoiding" with "avoids"

Page 2 line 32 and 33, suggest to add further references, suggest Rae, J. G. L. , G. AÃřalgeirsdóttir, T. Edwards, X. Fettweis, J. Gregory, H. Hewitt, J. Lowe, P. Lucas-Picher, R. Mottram, A. Payne, J. Ridley, S. Shannon, W. J. Van de Berg, R. Van de Wal, M. Van den Broeke. 2012. Greenland ice sheet surface mass balance: evaluating simulations and making projections with regional climate models, The Cryosphere, 6, 1275-1294, doi:10.5194/tc-6-1275-2012, 2012 - for line 33, as it compares several RCMs

Page 3, line 20 suggest to replace "their" with "the"

Page 3, line 24-25, this sentence is strange, perhaps replace "of being" with "to be"?

Page 5, line 8, add explanation of what H stands for

Page 6, line 27, is there missing "-1" on the unit for the decay parameter?

Page 6, line 28, suggest to add ",B," after Basal melt

Page 6, line 26, suggest to add ",W," after basal water layer

Page 7, line 6, "closest to the coordinates by Rignot and Mouginot (2012)" is no clear, suggest to edit to clarify what is meant here

Page 7, line 15 that statements "our method is already able to determine the subglacial discharge for each glacier" needs some quantification, how realistically do authors think the model results are?

Page 8, line 16, see comment above, is there something missing in the equation that uses the T from equation 9, to compute the surface ablation?

Page 8, line 20 and 28, see comment above, suggest to rename "implied flux" to something that has surface mass balance in the name

Page 9, line 10, suggest to rename "implied flux", see comment above

Page 9, line 24, suggest to add "in the 21st century"

Page 9, line 28, Why do you need forcing until 2300?

Page 10, line 14 how is the ice load changing? If the mass balance follows equation 10 the thickness is always kept close to the observed one?

Page 10, line 22 suggest to add reference for the "over-implicit ice thickness solver"

Page 10, in lines 21, 25 and 27 it is three times stated that the resolution is changed at 5 ky BP, suggest to edit to get rid of redundancy

Page 11, line 1, suggest to edit, it is not clear how many constants are tested, and at what time the comparison is made, is it at t=0?

Page 11, line 3, at what time is the comparison made?

Page 11, line 15, it is not clear what "from now on" means, after 1900?

Page 11, line 22, see comment above, what does "compares overall well" mean? At what time? Can it this comparison be quantified?

Page 11, line 24, see comment above, what does "slightly lower" mean, can it be quantified?

Page 11, line 25 see comment above, what does "somewhat smaller" mean?

Page 11, line 25-26, what does this sentence mean? Is the mismatch acceptable?

Page 11, line 27-32 these sentences are strange and not encouraging for the reader to accept the comparison, what does "fully resolved" (line 29) mean, and if you cannot model outlet glaciers with floating tounge, how is the NEGIS then the only large scale feature you cannot reproduce properly (what does "properly" mean here?) suggest to edit whole section

Page 12, line 3, what does "do not fully match he simulations" mean, can it be quantified?

Page 12, line 4, what does "rather smooth compared to observations" mean?

Page 12, line 6, what does "agrees basically" mean?

Page 12, line 17, see comment above about renaming "implied flux"

Page 12, line 18, what does "almost no change" mean?

Page 12 line 19, what does "a tiny volume change" mean? And "comparably small scale" ?

Page 12, line 32 strange beginning of sentence, suggest to edit (Certainly much stronger. . ..)

Page 13, line 6, what does "is minor" mean can it be quantified

Page 13, line 14, what does "much higher" mean?

Page 14, line 1, suggest to add "to" after "compared"

Page 14, line 8, what does "does not show big impacts" mean, can you quantify?

Page 14, line 24-25, strange sentence, it is not clear to what is being referred to, needs more explanation or clarification

Page 14, line 31-32, also here is strange sentence that needs clarification, deepening of basal topography by what?

Page 15, line 9, what does "an even smaller impact" mean?

Page 15, line 22, also here some quantification would be interesting "reasonably well" and "compared well" does not say enough about the success of the study - and line 33 "relatively large" does not give enough information

Page 16, line 5, what does "showed to be important" mean?

Page 17, lines 3 and 8, it is not clear what "horizontal time slices" mean, is this spatially distributed?

Page 17, line 8-12, this whole section is not clear, what is "favorable sampling interval"? and "not enough time slices" is not clear and "interval too long" also needs clarification

Page 17, line 14, what does this mean? How much overestimation?

Page 17, lines 20-21 the sentence "Note that the totals. . . " needs clarification

Page 25, table 2, Rows and columns have been put in wrong lines (interchange)

Figure 3, it is no clear at what time this comparison is made, would be useful to add that information to the figure caption

Figure 5, is 4b the difference between 5a and 5b? it would be useful to add third column with the difference and suggest to add boxes on 5C to show the location of smaller figures in Figure 6

Figure 6, suggest to add reference for observations, same as in Figure 5?

Figure 8, see comment above about renaming the term "implied flux"

Figure 9, is the green line, the total basal melt constant? Then give the exact number value in caption

---

## Referee Comment (RC2) · Anonymous Referee #2 · 14 May 2018

Calov et al. present new projections of Greenland melting contribution to sea level rise for 300 years into the future with their Greenland system model. They also estimate future evolution of sub-shelf basal melting rate for two outlet glaciers. Whilst the paper does not bring much of novelty it is nonetheless very nicely written and presents recent promising model developments. I therefore fully support its publication. I have however listed a few minor corrections/suggestions below that the authors should consider for their revised paper.

General comments:

- It is not clear, at least to me, how you deal with ocean-ice interaction. Does SICOPO-LIS simulate any ice shelves in your experiment? If yes, what do you choose as subshelf melting rate? Please make this clearer. Printer-friendly version

- Whilst the different model sub-components are generally well described I had difficulties in understanding what is the 1-D glacier model presented in Figure 1. My guess is that it corresponds to the coupler between the plume model and SICOPOLIS? Similarly as for your other arrows, you could add in Fig. 1 what is exchanged between the 1-D glacier model and the other components. In particular I do not understand what the right-to-left arrows stand for since the models are not interactively coupled yet.

Specific comments:

P3L29-30 Please show on a map where these two glaciers are located.

P6L7 How is defined the "shape of the glacier front"?

P7L8 "surface melt/runoff"  $\rightarrow$  It is not clear what this is. I assume it is the runoff provided by the MAR model (~rain minus retention due to refreezing?) and not surface ablation only?

P7L8-9 In your framework, you use the routing scheme of HYDRO (i.e. based on effective pressure) to route the water generated by surface melt. A fair amount of surface melt could be routed using surface gradient instead. Could you comment on how this can affect the pattern of discharge to the ocean?

P7L29-30 The gradients are not well defined for the accumulation regime because of precipitation that has a much more complex spatial pattern than temperature (and by extension ablation). I would guess that the vertical gradients for runoff are not well defined neither since a large part of runoff is composed by liquid precipitation. Could you comment on that?

P8L8-9 How is the surface temperature elevation gradient computed?

P8L25-27 A list of limitations of such an approach is welcome here, thanks. However I think you should expand more on the discussions of these. In particular, I think that taking into account the free-evolving topography during past cycles will have a large impact on simulated temperature profile as the ice thickness has considerably changed
during the last termination (Vinther et al., 2009) and the stress regime will be largely different with an ice sheet extended towards the continental shelf in glacial conditions. An other limitation of the SMB anomaly method to drive the spin-up is that an artificial SMB term is used to compensate all the model deficiencies in term of ice dynamics. A discussion on these points would be much appreciated.

P9L27-29 Please rephrase.

P10L9 Can't we expect a regional freshening due to the Greenland ice sheet melt? Why this could not be tested here as well with idealised scenarios (as for temperature)?

P10L22-24 The change in resolution certainly has an impact on the stress regime simulated by SICOPOLIS. Could you compare the state of your Greenland ice sheet (internal temperature) and your inferred surface mass balance (Mimpl) for present-day at 5 and 10 km resolution? Maybe you could add a few words on why doing the spin-up at a coarser resolution is not a problem in your case. It could also be interesting to have the future projections at 10 km resolution.

P10L25-26 Similarly to the change in resolution: have you tried to switch the hybrid mode before 500 years or after? How big is the impact on the simulated Greenland ice sheet? I assume that the thermal regime might be largely affected by the change in dynamics...

P11L4-5 Would it make sense to discuss the RMSE in SMB instead of the total difference in SMB?

P11L6-11 In the paper it could be worth discussing the spatial pattern of your difference in surface elevation and surface mass balance. From b-c we can really see that you have an important model drift at the margins: except for the NEGIS region your velocities seem too high (confirmed by Fig. 5) leading to negative surface elevation difference (compensated by artificial positive SMB anomaly). Related to this, why could you not find a Cb value minimising this model drift? TCD
P11L13-14 Do you need to spin-up HYDRO as well?

P11L24-25 The simulated surface velocity does not seem smaller to me when you look at the western flank of the ice sheet.

P11L26-27 Please reformulate.

P11L29-30 Why?

P12L18 There is no change in ice volume visible but it does not mean that the ice thickness is not changing as you have compensating errors.

P12L20-21 How this correction has been made? Do you use a point by point correction of ice thickness or do you simply use a correction of volume based on an averaged number? If the latter, how large would have been the difference when using the point by point correction?

P12L30 The effect is stronger with RCP8.5 when looking at the absolute value but relative changes are in fact smaller. Which is in agreement with Vizcaino et al. (2015).

P28 Fig. 3 Maybe you could add to these the evolution of total volume and RMSE of SMB for the different relaxation time.

P29 Fig. 4 Could you add more levels to your colour scale?

P31 Fig. 6 Do you have floating points? If yes, you should highlight them on this plot.

P32 Fig. 7 Could you comment on why you have sub-glacial lakes when the base is frozen?

Technical corrections:

P3L31 Dot instead of semicolon

P6L2 No capital S for "submarine"

P35 Fig. 10 Typo towards the end of the second line.
References

Vinther, B. M., S. L. Buchardt, H. B. Clausen, D. Dahl-Jensen, S. J. Johnsen, D. A. Fisher, R. M. Koerner, D. Raynaud, V. Lipenkov, K. K. Andersen, T. Blunier, S. O. Rasmussen, J. P. Steffensen and A. M. Svensson (2009), Holocene thinning of the Greenland ice sheet, Nature, 461, 385-388. doi: 10.1038/nature08355.

Vizcaino, M., U. Mikolajewicz, F. Ziemen, C. B. Rodehacke, R. Greve, and M. R.van den Broeke (2015), Coupled simulations of Greenland Ice Sheet and climate change up to A.D. 2300. Geophys. Res. Lett., 42, 3927-3935. doi: 10.1002/2014GL061142.

---

## Author Comment (AC1) · 19 Jun 2018

**Response to Reviewer 1**

We are grateful to the reviewer for the friendly and constructive review containing valuable remarks and suggestions for our present and future work. The suggestions will greatly improve our present paper.

The comments by the reviewer are in indented blocks and italic fonts.

**Response to specific comments**

> *The manuscript is well organized and clearly written, but I find the discussion of the results and comparison with data or other studies quite qualitative and think it would improve the paper if the authors would make more quantitative analysis of their results, see numerous comments below.*

Thank you very much for your positive opinion. In the revised version of our paper, we will quantify our analysis in all cases possibly.

> *The description of the forcing scheme is not clear (section 2.6) as it is not clear how the model applies the temperature defined in equation 8, as it seems that according to equation 10 the mass balance is only the difference between the modelled and measured elevation, divided by relaxation constant. Is this really the case, or is missing a description of a positive degree day method to compute the surface mass balance during the spin up period and current equation 10 would be one term of that forcing?*

The surface temperature defined in Eq. 8 applies directly to the surface of the ice sheet. Indeed, the first paragraph of this section is not fully clear. We will change the first paragraph by stating explicitly what our boundary conditions are. In our approach, the implied flux (now implied SMB, Eq. 10) is indeed only the difference between the observed and modelled surface elevation divided by a relaxation constant. We do not need any additional forcing in surface mass balance (SMB) for the palaeo runs, as the implied SMB flux (now named implied SMB) is the SMB that keeps the ice sheet on its observed shape, whereby the relaxation constant determines how close the model ice sheet is kept to the observed. The computation over the glacial cycle serves to yield the palaeo temperatures inside the ice sheet. The implied SMB at present-day is a by-product of this computation. In particular at the end of simulation, the implied SMB sustains the ice sheet near the observed present-day one. If there were not any model errors, the implied SMB would agree relatively well with the observed SMB. However, because there are model errors – in particular in areas where the model cannot resolve outlet glaciers – the implied SMB is nothing else than the observed SMB plus a correction of the errors of the ice sheet model. Please see also the figure, which we attached to our response to reviewer 2. We will add additional clarifications on this subject to Section 2.6.

> *The naming of the implied flux (equation 11) is confusing, suggest to call it something that indicates surface mass balance.*

This is good point. We will rename implied flux into implied surface mass balance, implied SMB.

> *The introduction section is comprehensive and gives a good overview of the current state of development of large scale ice sheet models, and there is an impressive reference list for this study. I find, however, that the first part of the introduction should have more references for the general statements (page 1, lines 21 and 22, as well as page 2 lines 1 and 2) or at least indicate that these are not the only papers stating those broad things, with "e.g." before that one reference.*

We will add at least five more new references, which we now found to the paragraph. After all, we cannot add all references. For the theme on accelerating mass loss, we consider to include an "e.g."

**Response to minor comments**

Thank you very much for all your comments. We will address all minor comments appropriately.

---

## Author Comment (AC2) · 19 Jun 2018

**Response to Reviewer 2**

We are grateful to the reviewer for the friendly and constructive review supporting publication of our paper. Your valuable remarks and suggestions will greatly improve our manuscript.

The comments by the reviewer are in indented blocks and italic fonts.

**Response to general comments**

> *It is not clear, at least to me, how you deal with ocean-ice interaction. Does SICOPOLIS simulate any ice shelves in your experiment? If yes, what do you choose as subshelf melting rate? Please make this clearer.*

In the model setup for this paper, SICOPOLIS does not simulate ice shelves and does not treat ice ocean interaction. Bi-directional ice ocean interaction was never planned in IGLOO, as can be seen in Fig.1. In the paper, changes in the ocean temperature are prescribed from data (Section 2.8) to the turbulent plume model. Impact of ocean temperature and subglacial ice discharge on submarine melting is examined in offline mode with the turbulent meltwater plume model. Concerning clarification about ice shelves, we will add more explanation to the paper.

> *Whilst the different model sub-components are generally well described I had difficulties in understanding what is the 1-D glacier model presented in Figure 1. My guess is that it corresponds to the coupler between the plume model and SICOPOLIS? Similarly as for your other arrows, you could add in Fig. 1 what is exchanged between the 1-D glacier model and the other components. In particular I do not understand what the right-to-left arrows stand for since the models are not interactively coupled yet.*

We will improve Fig. 1 by adding the variables for the submerge part of the outlet glaciers and for their submarine melt. Additionally, we will refer to Beckmann et al. (2018), who explains more details on the 1-D glacier and plume models, as parts of IGLOO. As noted in the paper, the coupling between the ice sheet and the outlet glaciers has not been implemented into IGLOO yet. To clarify this further, we will make the exchange arrows between the ice sheet and the outlet glaciers in Fig. 1 dashed now.

**Response to specific comments**

> *P3L29-30 Please show on a map where these two glaciers are located.*

We will add a new figure and will refer to it at the suggested place in the main text.

> *P6L7 How is defined the "shape of the glacier front"?*

We meant the submerged part of the outlet glacier and will change the term accordingly.

*P7L8 "surface melt/runoff" ! It is not clear what this is. I assume it is the runoff provided by the MAR model (_rain minus retention due to refreezing?) and not surface ablation only?*

Right and understood: melt is not synonym of runoff. It is the runoff, in our case from the MAR model. We will erase the word "melt".

*P7L8-9 In your framework, you use the routing scheme of HYDRO (i.e. based on effective pressure) to route the water generated by surface melt. A fair amount of surface melt could be routed using surface gradient instead. Could you comment on how this can affect the pattern of discharge to the ocean?*

We fully agree with you that the routing scheme in HYDRO can lead to different patterns of subglacial discharge reaching the ocean compared to a scheme, which uses surface gradients only. However, in case of complex bedrock with deeply incised structures, the routing scheme in HYDRO is more accurate. Apart from surface melt, there is also basal melt, which is important for winter and which our model accounts for.

*P7L29-30 The gradients are not well defined for the accumulation regime because of precipitation that has a much more complex spatial pattern than temperature (and by extension ablation). I would guess that the vertical gradients for runoff are not well defined neither since a large part of runoff is composed by liquid precipitation. Could you comment on that?*

The complete sentence reads: "For the surface mass balance, we apply the gradient method only to the ablation regime, because the regression is in many cases not well defined for the accumulation regime (Helsen et al., 2012)." We evaluated the data on surface mass from the MAR regional climate model and made the observation that the SMB gradients in the ablation area are much better defined compared to the gradients in the accumulation area. The issue can be seen in Fig. 2 by Helsen et al. (2012) too. The data in accumulation area appears noisier compared to the data in the ablation area. Therefore, it makes sense to determine a regression line solely in the ablation area, while in the accumulation area one can just assume zero gradients. We could add more explanation on this to the paper.

*P8L8-9 How is the surface temperature elevation gradient computed?*

We compute the surface temperature gradient via representative local gradients inside a search radius, as Helsen et al (2012) did for surface mass balance. This is in further detail explained in Section 2.5. We will improve Sections 2.5 and 2.6 and will better interlink between the sections relevant for this issue.

*P8L25-27 A list of limitations of such an approach is welcome here, thanks. However I think you should expand more on the discussions of these. In particular, I think that taking into account the free-evolving topography during past cycles will have a large impact on simulated temperature profile as the ice thickness has considerably changed during the last termination (Vinther et al.,*

*2009) and the stress regime will be largely different with an ice sheet extended towards the continental shelf in glacial conditions. An other limitation of the SMB anomaly method to drive the spin-up is that an artificial SMB term is used to compensate all the model deficiencies in terms of ice dynamics. A discussion on these points would be much appreciated.*

We will extent our discussion on the model limitations. The point on elevation changes during last glacial cycle is already in the list on page 8, lines 25-27. We will include Holocene in the discussion of model limitations. Concerning our method with implied SMB, we will add a paragraph to the discussion section, where we will discus the advantages (e.g. little drift due to a simulated surface elevation close to the observed surface elevation) and the deficiencies, which could be caused if the SMB correction becomes too strong. To point on this, our investigation of the impact of the relaxation constant on surface elevation and implied SMB inspects already the method. For the choice of the relaxation constant, we made a trade off to yield a surface elevation close to observations and at the same time make the correction via the implied SMB as small as possible. We will widen our discussion of the method.

*P9L27-29 Please rephrase.*

We will rephrase the sentence.

*P10L9 Can't we expect a regional freshening due to the Greenland ice sheet melt? Why this could not be tested here as well with idealised scenarios (as for temperature)?*

As IGLOO does not include interactive ocean, we are not able to test this hypothesis.

*P10L22-24 The change in resolution certainly has an impact on the stress regime simulated by SICOPOLIS. Could you compare the state of your Greenland ice sheet (internal temperature) and your inferred surface mass balance (Mimpl) for present-day at 5 and 10 km resolution? Maybe you could add a few words on why doing the spin-up at a coarser resolution is not a problem in your case. It could also be interesting to have the future projections at 10 km resolution.*

We needed the switch in the resolution to be able to perform the palaeo runs for model initialization within a reasonable computing time. We also needed many calculations to be able to adjust the model parameters. Such a switch is always a compromise between accuracy and computational time. To address your question on 5 and 10 km resolution, we attached a figure to the end of this response (Figure S1). As seen in Fig. S1, our fields with the switch to 5 km resolution show finer structures compared to the fields from the run in 10 km over the entire glacial cycle. In particular, the temperate basal regions have a much finer structure now and resolve much better the outlet glaciers and their catchment areas. It is evident that the base of outlet glaciers should be temperate in reality, what is resolved better in 5 km resolution in our model, even with the switch. For climate projection, we believe that the inaccuracy resulting from our switch in resolution (and the other switches) is small compared to the uncertainties in the scenarios and the

choice of GCM forcing. For the final paper if granted, we could further investigate the problem of resolution change. For sure, we will add discussion on this point to the paper. Concerning future projections in 10 km resolution, we think that this could be problematic, because the SMB data provided by MAR just are in 5 km resolution. For the future projections, this 5 km resolution is the best possible choice if we use the MAR data. A resolution of 10 km would be less accurate. For resolving the ablation area, resolution is especially critical. In lower resolution, we will loose grid points with negative SMB, because the ablation area is rather narrow. For the temperature field used for 10 km spin-up this is less problematic.

> *P10L25-26 Similarly to the change in resolution: have you tried to switch the hybrid mode before 500 years or after? How big is the impact on the simulated Greenland ice sheet? I assume that the thermal regime might be largely affected by the change in dynamics ...*

We did several computation concerning optimal points in time for the changes in regime. They led us to the conclusion that we need the last two switches (one for change in stress regime and another one for change from relaxation surface to free surface) to happen at different times, in order to avoid a model shock. On this issue, we will make investigations that are more systematic. Most probably, an earlier switch for change in stress regime will not make too much difference. Important to note, that we need at least these two switches, because we cannot effort to run the ice sheet model in the hybrid mode over the entire glacial cycle.

> *P11L4-5 Would it make sense to discuss the RMSE in SMB instead of the total difference in SMB?*

The choice of the measure is often arbitrarily. However, in our case, we used total difference in SMB, because we can compare it with total SMB from simulations with regional models of the Greenland ice sheet. Indeed, we use the difference in SMB for discussion in the discussion section.

> *P11L6-11 In the paper it could be worth discussing the spatial pattern of your difference in surface elevation and surface mass balance. From b-c we can really see that you have an important model drift at the margins: except for the NEGIS region your velocities seem too high (confirmed by Fig. 5) leading to negative surface elevation difference (compensated by artificial positive SMB anomaly). Related to this, why could you not find a Cb value minimising this model drift?*

We will add more discussion about our difference in surface elevation and surface mass balance to the paper. Indeed this is missing, as we solely focussed the text on the dependence of the field patterns on the relaxation time. Concerning impact of Cb value on implied SMB, we have not investigated this in very detail so far. As stated in the paper, we find our Cb value by minimizing the RMSE in surface velocity. We regard the surface velocity higher that 50 m/yr (as stated in paper) as the relevant measure for optimizing the basal sliding coefficient Cb, because sliding mostly affects fast flow. In our approach, we use MAR SMB and observed surface elevation to find an appropriate

relaxation constant (see Fig. 3). We think that this is reasonable, because the relaxation constant affects initial fields of surface elevation and implied SMB, which we use for the projections. Furthermore, with this method the total drift (measured in ice volume) is negligible, what is relevant for our sea level rise projections.

*P11L13-14 Do you need to spin-up HYDRO as well?*

HYDRO does not need any spin-up, as it is a diagnostic model. We will add a clarifying sentence on this matter to Section 2.2.

*P11L24-25 The simulated surface velocity does not seem smaller to me when you look at the western flank of the ice sheet.*

We meant the ridges and not the flanks. We will reformulate like: "The simulated surface velocities (smaller than 2 m/yr) along the ridges are somewhat smaller compared to the observed ones (often larger than 2 m/yr)". If necessary, we will further improve the paragraph.

*P11L26-27 Please reformulate.*

We will fix the sentence.

*P11L29-30 Why?*

In this paper, we operate SICOPOLIS in a setting, which does not treat ice shelves. We will clarify this.

*P12L18 There is no change in ice volume visible but it does not mean that the ice thickness is not changing as you have compensating errors.*

We are aware that in principle there can always be regional biases, which compensate each other in total. In our approach, the implied flux compensates particularly the regional errors of the ice sheet model. Indeed inspection of evolution of the ice thickness field could be used to refine our approach. We are grateful for this hint. In this paper, we use ice volume as indicator for the quality of our approach. As the change in ice volume in the run forced with implied flux only is very small, we regard ice volume as indicator as far sufficient.

*P12L20-21 How this correction has been made? Do you use a point by point correction of ice thickness or do you simply use a correction of volume based on an averaged number? If the latter, how large would have been the difference when using the point by point correction?*

We subtract the ice volume gained by the run forced with implied flux only from the ice volume resulting from the projection runs. This is done for every point in time. We will clarify this in the paper.

*P12L30 The effect is stronger with RCP8.5 when looking at the absolute value but relative changes are in fact smaller. Which is in agreement with Vizcaino et al. (2015).*

This is correct. Thank you very much for finding this. We will add some sentences about that.

*P28 Fig. 3 Maybe you could add to these the evolution of total volume and RMSE of SMB for the different relaxation time.*

As there is already a panel for the difference between the totals for simulated and observed SMB, this would not give to much new information. We would like to keep the figure as it is.

*P29 Fig. 4 Could you add more levels to your colour scale?*

We will add more levels to both colour bars.

*P31 Fig. 6 Do you have floating points? If yes, you should highlight them on this plot.*

In our model setting, the ice is restricted to land and cannot move into the ocean. Therefore, we do not have floating points.

*P32 Fig. 7 Could you comment on why you have sub-glacial lakes when the base is frozen?*

In our setting HYDRO computes the hydrological potential over the entire area of the ice sheet, although it is possible to restrict the computations to temperate basal regions. Such a restriction can lead to blocking effects and the basal water fluxes are hampered to reach the ocean in region where it should reach the ocean. If the hydrological potential is defined over the entire ice area, we are just on the safe side. Because we only allow basal sliding over temperate basal areas, it will make little difference to the ice dynamics. However, the basal water is more likely to reach its place in the ocean. By the way, our projected positions of present-day subglacial lakes correspond with findings by Livingstone et al. (2013). These positions are due to sinks in the hydrological potential. In this context, we would like to point out the great uncertainty in the determination of the thermal state of the Greenland basal ice (MacGregor et al., 2013).

**Technical corrections**

We will address the technical corrections appropriately.

**References**

Beckmann, J., Perrette, M., Beyer, S., Calov, R., Willeit, M., and Ganopolski, A.: Modeling the response of Greenland outlet glaciers to global warming using a coupled

flowline-plume model, The Cryosphere Discussions, pp. 1–32, doi:10.5194/tc-2018-89, in review, 2018.

**Figures**

[Figure]

*Figure S1: Present-day fields of implied SBM (panels a and b) and of basal temperature corrected for melting point (panels c and d). Left: 10 km × 10 km horizontal resolution over the entire glacial cycle. Right: Switching from 10 km × 10 km to 5 km × 5 km at 5 kyr BP.*

---

## Author Response (AR2)

**Response and Remarks to the Editor**

Both reviews were quite extensive. Therefore, we considerably revised the manuscript.

Although not complained by the reviewers, we change the usage of the term "climate sensitivity", as we were not precise with its usage. We meant "strength of the SMB forcing over Greenland" by "climate sensitivity".

Further, we included the Editor's technical corrections into the revised manuscript.

Our point-to-point response to the reviewers follow below. The full LaTeX-differences file is attached to the end of this document.

**Point to Point Response to all Reviewers**

The comments by the reviewers are in indented blocks and italic fonts.

**Response to Editor**

> *P18 L7 remove "somewhat".*

Done.

> *P18 L11 "is about zero" doesn't mean anything, I would suggest writing some qualitative statement like "smaller than MAR SMB" or "smaller than x% of MAR SMB".*

Done.

> *P18 L12 "reasonable small" doesn't mean anything. Again I would suggest rewrite the sentence with a qualitative statement.*

Done.

> *P18 L13 change "rather noisy" for noisier or similarly comparative statement.*

Done.

**Response to Reviewer 1**

**Response to specific comments of reviewer 1**

> *The manuscript is well organized and clearly written, but I find the discussion of the results and comparison with data or other studies quite qualitative and think it would improve the paper if the authors would make more quantitative analysis of their results, see numerous comments below.*

We quantified our analysis in all cases possibly. Details follow below.

> *The description of the forcing scheme is not clear (section 2.6) as it is not clear how the model applies the temperature defined in equation 8, as it seems that according to equation 10 the mass balance is only the difference between the modelled and measured elevation, divided by relaxation constant. Is this really the case, or is missing a description of a positive degree day method to compute the surface mass balance during the spin up period and current equation 10 would be one term of that forcing? 2*

The surface temperature defined in Eq. 8 applies directly to the surface of the ice sheet. Indeed, the first paragraph of this section is not fully clear.

We made considerable changes to this section. We changed the first paragraph of Section 2.6 (counting of Discussion paper, now Section 2.7), writing explicitly what are our boundary conditions for the palaeo run. To make it further clear we wrote which equations define our surface temperature and surface mass balance used for the palaeo simulations. Further, we now state explicitly that there is no additional forcing needed in Eq. (10), because this equation specifies an iteration for surface mass balance and surface elevation by running the ice sheet model in time.

> *The naming of the implied flux (equation 11) is confusing, suggest to call it something that indicates surface mass balance.*

We renamed "implied flux" into "implied SMB".

> *The introduction section is comprehensive and gives a good overview of the current state of development of large scale ice sheet models, and there is an impressive reference list for this study. I find, however, that the first part of the introduction should have more references for the general statements (page 1, lines 21 and 22, as well as page 2 lines 1 and 2) or at least indicate that these are not the only papers stating those broad things, with "e.g." before that one reference.*

We added five new references to the paragraph. For the theme on accelerating mass loss, we included an "e.g."

**Response to minor comments of reviewer 1**

> *Page 1 line 19 suggest to replace "melting" with "retreat"*

As we simulate submarine melting in our runs the term "melting" is correct here. No changes made to the manuscript.

> *Page 2 line 4-5, suggest to add something about the peripheral glaciers, that are not attached to the ice sheets, as those are actually contributing considerably to the sea level rise, see for example: Machguth, H., P. Rastner, T. Bolch, N. Mölg,*

*L. Sandberg Sørensen, G. AÃˇralgeirsdottir, J. H. van Angelen, M. R. van den Broeke and X. Fettweis. 2013. The future sea-level rise contribution of Greenland's glaciers and ice caps. Environ. Res. Lett. 8, 025005 doi:10.1088/1748-9326/8/2/025005*

The contribution of these glaciers sea level rise is small compared to the contribution of the ice sheet and their outlet glaciers. We added sentences on this topic to that paragraph and cite Forsberg et al. (2017) therein.

> *Page 2, line 6, suggest to add "a" before interplay*

We added an "an".

> *Page 2, line 8 suggest to replace "thought" with "intended"*

Done.

> *Page 2, line 10, suggest to replace "treating" with "treat"*

We replaced "treating" with "to treat".

> *Page 2, line 11 suggest to replace "fast" with something like "computationally effiecent"*

We rephrased to "computational efficiency".

> *Page 2 line 15, suggest to add references after "studies exist"*

The reference to the model studies follow already after the introductory sentence of the paragraph. There is no change necessary here.

> *Page 2, line 19, suggest to replace "can serve" with "serves"*

Done.

> *Page 2 line 20 suggest to delete "as" after "and"*

The word "as" is fine there, as it introduces a subordinate clause. No changes made.

> *Page 2 line 22 suggest to replace "elevation" with "ice thickness"*

Done.

> *Page 2 line 23 suggest to replace "elevation" with "ice thickness"*

Done.

> *Page 2 line 24 suggest to replace "due" with "according"*

Done.

*Page 2 line 25 suggest to add "the" before "projected"*

We think it correct, because we meant ice volume in general. No changes made.

*Page 2 line 27, suggest to delete "in" after "yielding" and add "of" before "the present day"*

We replaced "of yielding in" with "that it provides" and we added "of" before "the present day".

*Page 2, line 28 suggest to replace "avoiding" with "avoids"*

We replace "avoiding" with "prevents".

*Page 2 line 32 and 33, suggest to add further references, suggest Rae, J. G. L. , G. AÃˇralgeirsdóttir, T. Edwards, X. Fettweis, J. Gregory, H. Hewitt, J. Lowe, P. Lucas-Picher, R. Mottram, A. Payne, J. Ridley, S. Shannon, W. J. Van de Berg, R. Van de Wal, M. Van den Broeke. 2012. Greenland ice sheet surface mass balance: evaluating simulations and making projections with regional climate models, The Cryosphere, 6, 1275-1294, doi:10.5194/tc-6-1275-2012, 2012 - for line 33, as it compares several RCMs*

We added the reference.

*Page 3, line 20 suggest to replace "their" with "the"*

We rephrased the entire subordinate clause.

*Page 3, line 23-24, this sentence is strange, perhaps replace "of being" with "to be"?*

Right. We replaced "of being capable" with "to be suitable".

*Page 5, line 8, add explanation of what H stands for*

Done.

*Page 6, line 27, is there missing "-1" on the unit for the decay parameter?*

There was indeed a problem. We changed Eq. (7). The decay parameter is in the denominator now.

*Page 6, line 28, suggest to add ",B," after Basal melt*

Done.

*Page 6, line 26, suggest to add ",W," after basal water layer*

This makes no sense. "W" is already defined on page 6, line 22. No change made.

> *Page 7, line 6, "closest to the coordinates by Rignot and Mouginot (2012)" is no clear, suggest to edit to clarify what is meant here*

Rignot and Mouginot (2012) give geographical positions of outlet glaciers. We clarified this in the revised manuscript.

> *Page 7, line 15 that statements "our method is already able to determine the subglacial discharge for each glacier" needs some quantification, how realistically do authors think the model results are?*

We understand. That sentence has not too much meaning. We deleted the sentence. The section describes the coupling of SICOPLOIS-HYDRO with the plume model. At this place, there is nothing to quantify. Our results on this are already quantified in the results section of the original paper and are further quantified in the revised version of the paper. See below.

> *Page 8, line 16, see comment above, is there something missing in the equation that uses the T from equation 9, to compute the surface ablation?*

Equation 10 is correct. No further input variables are necessary therein. We addressed the issue in our response to specific comments of reviewer 1 already. We improved Section 2.6 (Discussion paper counting, Section 2.7 revised manuscript) considerably.

> *Page 8, line 20 and 28, see comment above, suggest to rename "implied flux" to something that has surface mass balance in the name*

Done. We use the term implied SMB now.

> *Page 9, line 10, suggest to rename "implied flux", see comment above*

Done. We use the term implied SMB now.

> *Page 9, line 24, suggest to add "in the 21st century"*

Done.

> *Page 9, line 28, Why do you need forcing until 2300?*

In other papers, projections beyond 2100 (until the year 2200 or 2300) are often done in context of ice sheet modelling. This way we can compare with the other publications, what we do in the paper.

We rephrased the sentence and explained why we extended the forcing to year 2300.

> *Page 10, line 14 how is the ice load changing? If the mass balance follows equation 10 the thickness is always kept close to the observed one?*

This is correct. The change in ice load is minor. We opted for free bedrock, because this is a standard setup for SICOPOLIS palaeo runs. We are aware that the change in bedrock is minor if the ice surface relaxes to observed. Nothing is wrong with such setup.

Now we write: "Using a standard setting of SICOPOLIS for palaeo runs, …"

> *Page 12, line 18, what does "almost no change" mean?*

We deleted that sentence and quantified the drift.

> *Page 12 line 19, what does "a tiny volume change" mean? And "comparably small scale" ?*

The point here is that the drift is small compared with the sea level change in the projections. We made this clearer now. Additionally, we improved the quantification of our model drift.

> *Page 12, line 32 strange beginning of sentence, suggest to edit (Certainly much stronger. …)*

We rephrased the sentence.

> *Page 13, line 6, what does "is minor" mean can it be quantified*

Right. This sentence is about results we have not shown in the paper. Consequently, we erased the entire sentence now.

> *Page 13, line 14, what does "much higher" mean?*

For mass of the ice sheet the effect amounts in at the end of 2100 AD 150 Gt/yr and 290 Gt/yr at the end of 2300 AD. We added a sentence for quantification.

> *Page 14, line 1, suggest to add "to" after "compared"*

Right. Done.

> *Page 14, line 8, what does "does not show big impacts" mean, can you quantify?*

At the end of 2100 AD, the effect increases the annual submarine melt rate of Helheim Glacier and Store Glacier by 76 m/yr and 81 m/yr, respectively. We added some sentences for detailed quantification.

> *Page 14, line 24-25, strange sentence, it is not clear to what is being referred to, needs more explanation or clarification*

Right. We added more explanation and clarification to that paragraph.

> *Page 14, line 31-32, also here is strange sentence that needs clarification, deepening of basal topography by what?*

We deleted that sentence.

> *Page 15, line 9, what does "an even smaller impact" mean?*

Right. We quantified this by comparing our results with the results by Edwards et al. (2014).

> *Page 15, line 22, also here some quantification would be interesting "reasonably well" and "compared well" does not say enough about the success of the study - and line 33 "relatively large" does not give enough information*

First point (Page 15, line 22): We moved the first sentence of that paragraph to the first paragraph of the conclusions Section. We deleted the remainder of that paragraph. Second point: We gave an RMS error for "very large" there and added a reference to Fig. 3a (counting of discussion paper, Fig 5a revised paper).

> *Page 16, line 5, what does "showed to be important" mean?*

We rephrased the sentence.

> *Page 17, lines 3 and 8, it is not clear what "horizontal time slices" mean, is this spatially distributed?*

Specified: Now we use the term MAR fields (the longitudinal-latitudinal distributed fields for annual mean of surface temperature, SMB and monthly surface runoff for a given year simulated by MAR).

> *Page 17, line 8-12, this whole section is not clear, what is "favorable sampling interval"? and "not enough time slices" is not clear and "interval too long" also needs clarification*

We considerably rewrote the entire section. We added a new figure to that section for illustration (Fig A1).

> *Page 17, line 14, what does this mean? How much overestimation?*

Same as before: we rewrote the entire section.

> *Page 17, lines 20-21 the sentence "Note that the totals… " needs clarification*

Same as before: we rewrote the entire section.

> *Page 25, table 2, Rows and columns have been put in wrong lines (interchange)*

The caption of Table 2 was wrong. Therein, we changed "columns" with "rows" in the figure caption.

> *Figure 3, it is no clear at what time this comparison is made, would be useful to add that information to the figure caption*

It is for present day. We added that information to the figure caption (now Fig. 5).

> *Figure 5, is 4b the difference between 5a and 5b? it would be useful to add third column with the difference and suggest to add boxes on 5C to show the location of smaller figures in Figure 6*

First point: Figure 4 (Discussion paper counting, now Fig. 6) is from SIA runs in 10 km resolution, while Fig. 5 (now Fig. 7) is from a hybrid run in 5 km resolution. We explicitly say this in the paper. Fig. 4 (now Fig. 6) was to illustrate the dependence of the model on the relaxation time. Fig. 5 (now Fig. 7) shows the resulting elevation with an optimal relaxation. Differences in simulated elevation to observed one are rather small. Moreover, as we already zoom in to velocity in Fig. 6 (now Fig. 8), we do not think that it makes sense to discus more details in a differences plot. Second point: We added rectangles to Fig. 5c (now Fig. 7c) to indicate the position of the enlarged areas in Fig. 6 (now Fig. 8). We changed the figure captions accordingly.

> *Figure 6, suggest to add reference for observations, same as in Figure 5?*

We added the reference to the caption of Fig. 6 (now Fig. 8).

> *Figure 8, see comment above about renaming the term "implied flux"*

Right. We change the inset of the figure: "IMPL" is changed to "iSMB". We changed the figure caption: "implied flux" is changed to "implied SMB (iSMB)". Now Fig. 10.

> *Figure 9, is the green line, the total basal melt constant? Then give the exact number value in caption*

We intended to show that the total basal melt is small compared to the total surface runoff. Furthermore, the total basal melt is nearly constant and range between 10 and 12 Gt/yr. We found that our 15 Gt/yr were wrong after checking the model output. We added clarification to the main text and the figure caption for Fig. 9 (now Fig. 11).

**Response to Reviewer 2**

**Response to general comments of reviewer 2**

> *It is not clear, at least to me, how you deal with ocean-ice interaction. Does SICOPOLIS simulate any ice shelves in your experiment? If yes, what do you choose as subshelf melting rate? Please make this clearer.*

In the model setup for this paper, SICOPOLIS does not simulate ice shelves. We included a new subsection (Section 2.1 "Overview of IGLOO"). This subsection clarifies the point with the ice shelves.

> *Whilst the different model sub-components are generally well described I had difficulties in understanding what is the 1-D glacier model presented in Figure 1.*

*My guess is that it corresponds to the coupler between the plume model and SICOPOLIS? Similarly as for your other arrows, you could add in Fig. 1 what is exchanged between the 1-D glacier model and the other components. In particular I do not understand what the right-to-left arrows stand for since the models are not interactively coupled yet.*

We added a new subsection (Section 2.1 "Overview of IGLOO") to the beginning Section 2 to specify the used model components of IGLOO. Further, we improved Fig. 1 by adding the variables for the submerge part of the outlet glaciers and for their submarine melt. The exchange arrows between the ice sheet and the outlet glaciers in Fig. 1 are dashed now. Additionally, we refer to Beckmann et al. (2018), who explains more details on the glacier and plume models.

**Response to specific comments of reviewer 2**

*P3L29-30 Please show on a map where these two glaciers are located.*

We added a new figure (Fig. 2) with a map, which shows the location of the two glaciers.

*P6L7 How is defined the "shape of the glacier front"?*

We meant the submerged part of the outlet glacier and changed the term accordingly.

*P7L8 "surface melt/runoff" ! It is not clear what this is. I assume it is the runoff provided by the MAR model (_rain minus retention due to refreezing?) and not surface ablation only?*

Understood: melt is not synonym of runoff. It is the runoff, in our case from the MAR model. Due to the demands of reviewer 1, we erased the entire sentence. However, in the revised version of the paper, we no more use the term "melt/runoff".

*P7L8-9 In your framework, you use the routing scheme of HYDRO (i.e. based on effective pressure) to route the water generated by surface melt. A fair amount of surface melt could be routed using surface gradient instead. Could you comment on how this can affect the pattern of discharge to the ocean?*

We commented to this in our online response: We fully agree with the reviewer that the routing scheme in HYDRO can lead to different patterns of subglacial discharge reaching the ocean compared to a scheme, which uses surface gradients only. However, in case of complex bedrock with deeply incised structures, the routing scheme in HYDRO is more accurate. Apart from surface melt, there is also basal melt, which is important for winter and which our model accounts for.

We made no changes in the revised manuscript on that, because we have not made calculations with routing via surface gradient. Therefore, we cannot quantify this effect.

*P7L29-30 The gradients are not well defined for the accumulation regime because of precipitation that has a much more complex spatial pattern than*

*temperature (and by extension ablation). I would guess that the vertical gradients for runoff are not well defined neither since a large part of runoff is composed by liquid precipitation. Could you comment on that?*

Response: We evaluated the data on surface mass from the MAR regional climate model and made the observation that the SMB gradients in the ablation area are much better defined compared to the gradients in the accumulation area. The issue can be seen in Fig. 2 by Helsen et al. (2012) too. The data in the accumulation area appears noisier compared to the data in the ablation area. Therefore, it makes sense to determine a regression line solely in the ablation area, while in the accumulation area one can just assume zero gradients.

Changes to the manuscript: We added some more explanation on this to that section.

*P8L8-9 How is the surface temperature elevation gradient computed?*

We compute the surface temperature gradient via representative local gradients inside a search radius, as Helsen et al (2012) did for surface mass balance. This is in further detail explained in Section 2.5 (Section 2.6 in the revised paper).

Changes made in the manuscript: We improved Section 2.5 (now Section 2.6) to make more visible that the surface temperature gradient is determined analogous to the SMB gradient. In Section 2.6 (now Section 2.7) and Section 2.7 (now Section 2.8) we refer to Section 2.5 (now Section 2.6) concerning the gradients now.

*P8L25-27 A list of limitations of such an approach is welcome here, thanks. However I think you should expand more on the discussions of these. In particular, I think that taking into account the free-evolving topography during past cycles will have a large impact on simulated temperature profile as the ice thickness has considerably changed during the last termination (Vinther et al., 2009) and the stress regime will be largely different with an ice sheet extended towards the continental shelf in glacial conditions. An other limitation of the SMB anomaly method to drive the spin-up is that an artificial SMB term is used to compensate all the model deficiencies in terms of ice dynamics. A discussion on these points would be much appreciated.*

Response: Such a discussion makes sense.

Changes made: We moved the paragraph describing the limitation of the relaxation approach to the discussion section (Section 6), where we extended considerably our discussion on the model limitations.

*P9L27-29 Please rephrase.*

We considerably improved that paragraph.

*P10L9 Can't we expect a regional freshening due to the Greenland ice sheet melt? Why this could not be tested here as well with idealised scenarios (as for temperature)?*

As IGLOO does not include interactive ocean, we are not able to test this hypothesis.

*P10L22-24 The change in resolution certainly has an impact on the stress regime simulated by SICOPOLIS. Could you compare the state of your Greenland ice sheet (internal temperature) and your inferred surface mass balance (Mimpl) for present-day at 5 and 10 km resolution? Maybe you could add a few words on why doing the spin-up at a coarser resolution is not a problem in your case. It could also be interesting to have the future projections at 10 km resolution.*

We commented to this in our online response and included a figure therein. The simulation with the resolution switch already shows a finer structure of the basal temperature field compared to that with lower resolution.

Changes to the manuscript: We added a new paragraph on the first switch to the section on initialization.

*P10L25-26 Similarly to the change in resolution: have you tried to switch the hybrid mode before 500 years or after? How big is the impact on the simulated Greenland ice sheet? I assume that the thermal regime might be largely affected by the change in dynamics ...*

We commented to this in our online response. Mainly, we promised to investigate this in more detail, what did for the revised manuscript. We made a test calculation, which investigates the switch from shallow ice approximation to hybrid.

Changes in the manuscript: We added a new paragraph and a new figure on the second and third switch to Section 3 (Model initialization via palaeo runs).

*P11L4-5 Would it make sense to discuss the RMSE in SMB instead of the total difference in SMB?*

Response: The choice of the measure is often arbitrarily. However, in our case, we used the total difference in SMB, because we can compare it with the total SMB from simulations with regional models of the Greenland ice sheet. Indeed, we use the difference in SMB for discussion in the discussion section.

No changes made to the manuscript.

*P11L6-11 In the paper it could be worth discussing the spatial pattern of your difference in surface elevation and surface mass balance. From b-c we can really see that you have an important model drift at the margins: except for the NEGIS region your velocities seem too high (confirmed by Fig. 5) leading to negative surface elevation difference (compensated by artificial positive SMB anomaly). Related to this, why could you not find a Cb value minimising this model drift?*

Indeed this is missing, as we solely focussed the text on the dependence of the field patterns on the relaxation time.

Changes made: We discuss the elevation differences and how they relate to implied SMB and velocity pattern now. Concerning proposal by the referee to use Cb to minimize surface elevation difference, we commented to this in our online response.

> *P11L13-14 Do you need to spin-up HYDRO as well?*

HYDRO does not need any spin-up, as it is a diagnostic model. We added a clarifying sentence on this matter to the first paragraph of Section 2.2 (Section 2.3 revised manuscript).

> *P11L24-25 The simulated surface velocity does not seem smaller to me when you look at the western flank of the ice sheet.*

We wrote about the ridges and not about the flanks. We reformulated the sentence.

> *P11L26-27 Please reformulate.*

We fixed the sentence.

> *P11L29-30 Why?*

In this paper, we operate SICOPOLIS in a setting, which does not treat ice shelves. We clarified this.

> *P12L18 There is no change in ice volume visible but it does not mean that the ice thickness is not changing as you have compensating errors.*

We commented to this in our online response: We are aware that in principle there can always be regional biases, which compensate each other in total. In our approach, the implied flux compensates particularly the regional errors of the ice sheet model. Indeed inspection of evolution of the ice thickness field could be used to refine our approach. We are grateful for this hint. In this paper, we use ice volume as indicator for the quality of our approach. As the change in ice volume in the run forced with implied flux only is very small, we regard ice volume as indicator as far sufficient.

 No changes made to the manuscript.

> *P12L20-21 How this correction has been made? Do you use a point by point correction of ice thickness or do you simply use a correction of volume based on an averaged number? If the latter, how large would have been the difference when using the point by point correction?*

We subtract the ice volume gained by the run forced with implied flux only from the ice volume resulting from the projection runs. This is done for every point in time. We clarified this in Section 5.1.

*P12L30 The effect is stronger with RCP8.5 when looking at the absolute value but relative changes are in fact smaller. Which is in agreement with Vizcaino et al. (2015).*

This is correct. We added some sentences about the relative impact of this feedback to Section 5.1.

*P28 Fig. 3 Maybe you could add to these the evolution of total volume and RMSE of SMB for the different relaxation time.*

As there is already a panel for the difference between the totals for simulated and observed SMB, this would not give to much new information. We would like to keep the figure as it is.

*P29 Fig. 4 Could you add more levels to your colour scale?*

We added more levels to both colour bars (now Fig. 6).

*P31 Fig. 6 Do you have floating points? If yes, you should highlight them on this plot.*

In our model setting, the ice is restricted to land and cannot move into the ocean. Therefore, we do not have floating points. This means that there is not need for changes in the manuscript.

*P32 Fig. 7 Could you comment on why you have sub-glacial lakes when the base is frozen?*

We commented to this in our online response and added several sentences on this to the end of Section 4.

**Response to technical corrections of reviewer 2**

*P3L31 Dot instead of semicolon*

Done.

*P6L2 No capital S for "submarine"*

Done.

*P35 Fig. 10 Typo towards the end of the second line.*

Indeed, there was a problem with the sentence. We simplified the sentence.

**References**

[revised manuscript text omitted]